# Long-term monitoring of two endangered freshwater mussels (Bivalvia: Unionidae) reveals how demographic vital rates are influenced by species life history traits

Tim Lane[1], Jess Jones[2]*, Brett Ostby[3], Robert Butler[4]

1 Virginia Department of Wildlife Resources, Aquatic Wildlife Conservation Center, Marion, Virginia, United States of America, 2 Department of Fish and Wildlife Conservation, U.S. Fish and Wildlife Service, Virginia Polytechnic Institute and State University, Blacksburg, Virginia, United States of America, 3 Daguna Consulting, LLC Bristol, Bristol, Virginia, United States of America, 4 U.S. Fish and Wildlife Service, Asheville, North Carolina, United States of America

* Jess_Jones@fws.gov

**Data Availability Statement:** All relevant data are within the manuscript and its Supporting Information files.

## Abstract

To meet monitoring and recovery planning needs, demographic vital rates of two endangered freshwater mussels (Bivalvia: Unionidae)—the Cumberlandian Combshell (*Epioblasma brevidens*, Lea 1831) and Oyster Mussel (*Epioblasma capsaeformis*, Lea 1834), species endemic to the Tennessee and Cumberland river basins, U.S.A—were estimated and compared using census methodologies. Annual variation in population density and size, recruitment rate, mortality rate, sex ratios, and female fecundity of both species were observed from 2004–2014 at three fixed sites, spanning a 33.8 kilometer (KM) reach of the Clinch River, Hancock County, Tennessee. Mean population size of *E. brevidens* estimated from 11 censuses was 2,598 individuals at Swan Island (KM 277.1), 8,744 at Frost Ford (KM 291.8), and 879 at Wallen Bend (KM 309.6); collectively, these demes grew at an annual rate of 7% over the study period. Mean population size of *E. capsaeformis* was 7,846 individuals at Swan Island, 265,442 at Frost Ford, and 11,704 at Wallen Bend; collectively, these demes grew at an annual rate of 6%. Population size, variability in population growth, recruitment, and mortality of the shorter-lived *E. capsaeformis* (maximum age = 16 yrs, rarely >10 yrs) were higher than those of the longer-lived *E. brevidens* (maximum age = 25 yrs). Stream discharge was associated with realized per-capita population growth rate for both species when juvenile (Ages 1–3) data was included. Linear regression analysis showed that the growth rate of *E. brevidens* was negatively associated with median annual discharge ($p = 0.0274$) and that growth rate of *E. capsaeformis* was negatively associated with the number of days having extreme high discharge preceding a census ($p = 0.0381$). Fecundity of female *E. brevidens* averaged 34,947 (SE = 2,492) glochidia and ranged from 18,987 to 56,151, whereas fecundity of female *E. capsaeformis* averaged 9,558 (SE = 603) glochidia and ranged from 3,456 to 22,182. Estimated vital rates indicated that the two species are characterized by different life-history strategies, with *E. brevidens* exhibiting a periodic strategy (between *K*- and *r*-selected) and *E. capsaeformis* an opportunistic strategy (*r*-selected). These life history strategies are likely influenced by each species' longevity and

**Funding:** The study was funded by the U.S. Fish and Wildlife Service (USFWS), Asheville North Carolina Field Office (https://www.fws.gov/asheville/). The grant was received as a U.S. Geological Survey Research Work Order (RWO#140 to the Virginia Cooperative Research Unit) by JWJ as the primary investigator. Daguna Consulting, LLC provided support in the form of salary for author BJKO. The specific roles of the authors are articulated in the 'author contributions' section. The funders had no role in study design, data collection and analysis, decision to publish, or preparation of the manuscript.

**Competing interests:** The authors have read the journal's policy and the authors of this manuscript have the following competing interests: BJKO is a paid employee of Daguna Consulting, LLC. There are no patents, products in development or marketed products associated with this research to declare. This does not alter our adherence to PLOS ONE policies on sharing data and materials.

habitat preference, in addition to the life histories and population dynamics of their primary fish hosts.

# Introduction

The Clinch River upstream of Norris Reservoir in Hancock County, Tennessee (TN), supports 48 extant mussel species, including 20 listed as federally endangered [1–3]. This mussel assemblage in the TN section of the river is one of the more robust in the United States and offers an opportunity to monitor populations of rare and endangered species that are now extirpated throughout most of their ranges. Two species in particular, Cumberlandian Combshell (*Epioblasma brevidens*) and Oyster Mussel (*E. capsaeformis*), are so rare outside the Clinch River, that the collection of quantitative demographic data is nearly impossible elsewhere. Several studies have established long-term monitoring sites in Clinch River spanning the largest extant populations of *E. brevidens* and *E. capsaeformis* remaining, offering a unique opportunity to the mussel conservation community to simultaneously monitor multiple rare species under stable and natural conditions [1–4]. This has allowed managers to explore demographic fluctuations and the degree to which they are driven by environmental and community-level dynamics.

Demographic data help to address the monitoring component of Strategic Habitat Conservation (SHC), developed by the U.S. Fish and Wildlife Service (USFWS), and other components of this comprehensive approach to habitat management and conservation [5]. Further, the federal recovery plan for both species identifies collection of demographic data as a critical component for assessing the recovery of each species [6]. More broadly geographically, collection of demographic data is essential for evaluating success of mussel reintroductions and augmentations of these and other species throughout the Tennessee-Cumberland River Zoogeographic Province (TCRP). Rivers in the TCRP (e.g., Clinch, Powell, Nolichucky, French Broad, Paint Rock, Duck, Big South Fork Cumberland) harbor >30 endangered mussel species, and are considered the most diverse streams in the Appalachian Landscape Conservation Cooperative (LCC), which contains more imperiled mussels than any other LCC [7]. Collectively, these rivers are high-priority areas for reintroduction of many of these endangered or imperiled mussels in the "Plan for the Population Restoration of Freshwater Mollusks of the Cumberlandian Region" [8] and in numerous USFWS recovery plans. While mussel propagation and stocking efforts are advancing quickly in the region, advances in demographic assessment have lagged. Hence, there is a need to obtain statistically defensible estimates of parameters including but not limited to populations size, population growth rate, recruitment, fecundity, and age-specific survival [9]. These parameter estimates are inputs for population viability analyses (PVAs) and are critical for assessing consequences of climate change and other anthropogenic disturbances. It is anticipated results from this study will be applicable to other regions of the country, and thus help guide management of other mussel species.

Typical of most unionids, the life cycles of *E. brevidens* and *E. capsaeformis* include an obligate parasitic larval stage (glochidia), which females release for encystment on host fish. Encystment on the gill or body is required for the glochidia to transform into a free-living juvenile (4). Females of *E. brevidens* and *E. capsaeformis* have evolved highly specialized mantle lures to attract their respective fish hosts. For example, female *E. brevidens* have 2–4 small vesicles resembling fish eggs adjacent to a group of short papillae and recurving denticles on both valves that are used to capture and hold onto large *Percina* spp., e.g., *P. burtoni*, *P. aurantiaca*,

and *P. caprodes*. Females of *E. capsaeformis* have a brightly colored and broadly expanded mantle lure that, 1) provides strong contrast to a mobile macroinvertebrate-mimicking lure positioned on the mantle edge, and 2) inflates to immobilize *Etheostoma* spp., e.g., *E. rufilineatum*, *E. caureulem*, *E. zonale*, with the aid of small denticles along the outside of the marsupial expansion [10,11]. Each species is considered to be a host specialist, such that transformation success is only possible on a particular set of fish hosts. Habitat preferences for freshwater mussels is in part influenced by habitat preferences and availability of their host fishes, although little research has been conducted to test this assumption (Strayer 2008). Larger-bodied darters such as species in the genus *Percina* are known to migrate over greater distances than previously thought, with average movement >10 km and even as far as 55 km [12].

The purpose of this study was to provide estimates of demographic parameters for *E. brevidens* and *E. capsaeformis* to guide recovery and PVA [13] of each species in the TCRP. The primary objectives of this study were to: (1) quantitatively assess variation in density and population size of *E. brevidens* and *E. capsaeformis* in the Clinch River from 2004–2014; (2) estimate demographic vital rates, including sex-ratios, fecundity, recruitment, mortality, and population growth; and (3) test how stream discharge may influence these demographic parameters. Secondarily, this study sought to understand causes of variation in these estimates, and finally, explore how parameter estimates elucidate life history traits by comparing and contrasting *E. brevidens* with *E. capsaeformis*, specifically assessing the degree to which these species conform to previously identified life history strategies.

## Materials and methods

### Study area and site selection

The study area was a 33.8 kilometer (km) reach of the Clinch River extending from river kilometer (RKM) 276 upstream to RKM 309.8 in Hancock County, TN, which is a sub-set of previously assessed monitoring sites [1–4] (Fig 1). The typical habitat of *Epioblasma brevidens* and *E. capsaeformis* is shallow riffles and runs <1 meter (m) deep in stream areas composed of stable gravel/cobble substrate. This type of habitat (shoals) is abundant in the river but is interspersed with longer, slower-flowing runs and pools (>1 RKM), containing lower quality mussel habitat. Typical lengths of shoals in this river reach are 100–200 m long but occasionally longer [2]. All field work sampling for this project was conducted under permit #963 granted from Tennessee Wildlife Resources Agency.

### Population demographics

Mussel population densities were quantitatively assessed by quadrat excavation at three long-term monitoring sites in the Clinch River, including Swan Island (SI), RKM 277.1 [river mile (RM) 172.2]; Frost Ford (FF) RKM 291.8 (RM 181.3); and Wallen Bend (WB) RKM 309.6 (RM 192.4) (Fig 1; Tables 1 and 2). Each site was annually monitored from 2004–2014 for mussel community composition, species population size, and demographic trends [2–4,13], encompassing 10 time intervals between the 11 censuses. Sites were selected to represent the upper, middle, and lower reaches of the TN section of the river where healthy populations of numerous rare and endangered mussel species remain. Sampling was conducted in late summer and early autumn (September–October) when water levels were low, and young-of-the-year mussels had reached sizes large enough for detection (e.g., >10 mm). Upstream and downstream boundaries of sampling sites were discrete in the river, and had been previously determined by visual inspection of substrate composition (e.g., noting an abrupt change from suitable sand/gravel/cobble substrate to unsuitable bedrock or unstable soft sediments), water depth, flow velocity, and general absence of mussels [2]. Each year, the same upstream and

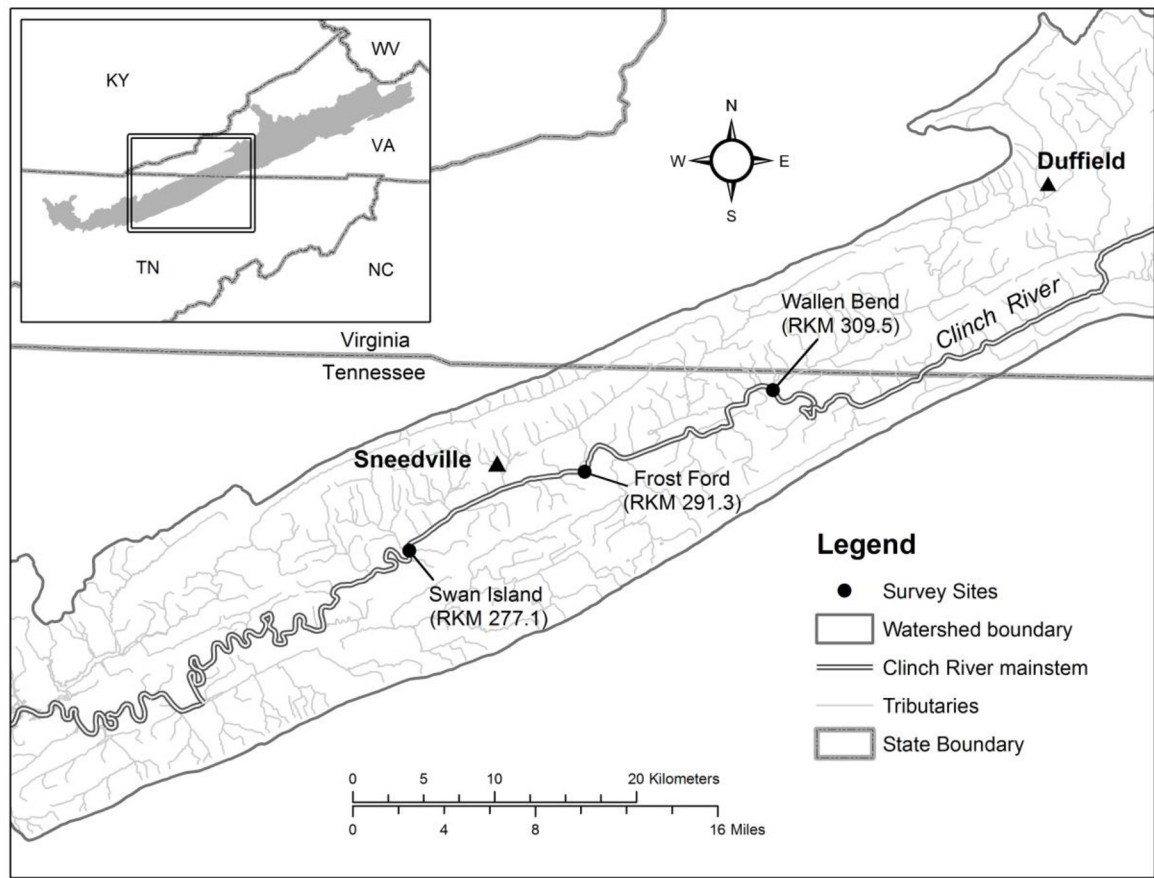

**Fig 1.** *Study area–annual monitoring sites in the Clinch River, Hancock County, TN, where populations of Epioblasma brevidens and E. capsaeformis were quantitatively surveyed to estimate demographic parameters from 2004–2014; sites included Swan Island, Frost Ford, and Wallen Bend.* Kyles Ford was used to collect gravid female mussels of both species to estimate fecundity.

downstream boundaries were used at sites. Site dimensions (length and width) were measured using a laser rangefinder (R400 HALO 6 X 24, Wildgame Innovations Inc., Broussard, Louisiana). Quantitative data were collected by systematic ¼ m$^2$ quadrat samples on transect lines placed perpendicular and equidistant along the width of the river. Quadrats were evenly spaced along transects, starting from a randomly assigned point between 0 and 10 meters from the right ascending bank (RAB). Total area (m$^2$) of the sample sites was determined by multiplying mean river width, measured at 10–20 m intervals, by total length of the reach. The total number of quadrats excavated per site and from year-to-year varied from 60–90 samples in past census years [2,3], so at least 80 quadrat samples were excavated at each site from 2012–2014 to increase precision of estimated mussel densities (Table 1). Extra quadrat samples were added across the original transect lines configured per site by simply adding additional evenly spaced samples per transect. Regardless of actual number, all quadrats were distributed using the same unbiased spatial distribution within well-defined site boundaries, making comparison among years valid. Using a mask and snorkel, surveyors visually searched for mussels while excavating the riverbed substrate to a depth of approximately 20 cm in each quadrat. Live mussels were collected, placed in a mesh bag, and brought to the riverbank for identification and measurement. Mussels were identified to species, sexed, measured for total shell

**Table 1. River kilometer (RKM) and river mile (RM) for each site censused from 2004–2014 in the Clinch River, TN.** The total number of ¼ m² quadrats excavated and raw number of individuals encountered in quadrats were used to estimate mean densities (mussels/m²) and 95% confidence limits (CL) for *Epioblasma brevidens* and *E. capsaeformis* by census year.

| Site name RKM, RM | Years sampled | No. ¼ m² quadrats yr⁻¹ | *Epioblasma brevidens* | | | | *Epioblasma capsaeformis* | | | |
|---|---|---|---|---|---|---|---|---|---|---|
| | | | No. individuals in quadrats (N) | Mean density ($\hat{D}$) m⁻² (SE) | Lower 95% CL | Upper 95% CL | No. individuals in quadrats (N) | Mean density ($\hat{D}$) m⁻² (SE) | Lower 95% CL | Upper 95% CL |
| Swan Island RKM 277.1 RM 172.2 | 2004 | 60 | 6 | 0.40 (0.02) | 0.18 | 0.87 | 10 | 0.67 (0.03) | 0.35 | 1.27 |
| | 2005 | 60 | 2 | 0.13 (0.01) | 0.03 | 0.54 | 9 | 0.60 (0.03) | 0.30 | 1.20 |
| | 2006 | 60 | 3 | 0.20 (0.02) | 0.06 | 0.62 | 2 | 0.13 (0.01) | 0.03 | 0.54 |
| | 2007 | 60 | 9 | 0.60 (0.03) | 0.30 | 1.20 | 19 | 1.27 (0.04) | 0.80 | 2.01 |
| | 2008 | 60 | 4 | 0.27 (0.02) | 0.08 | 0.89 | 11 | 0.73 (0.03) | 0.38 | 1.42 |
| | 2009 | 72 | 13 | 0.72 (0.03) | 0.36 | 1.46 | 27 | 1.50 (0.04) | 0.97 | 2.33 |
| | 2010 | 60 | 6 | 0.40 (0.02) | 0.18 | 0.87 | 28 | 1.87 (0.05) | 1.24 | 2.82 |
| | 2011 | 90 | 20 | 0.89 (0.03) | 0.48 | 1.64 | 51 | 2.27 (0.04) | 1.57 | 3.27 |
| | 2012 | 80 | 16 | 0.80 (0.03) | 0.42 | 1.53 | 70 | 3.50 (0.07) | 2.51 | 4.89 |
| | 2013 | 80 | 7 | 0.35 (0.01) | 0.17 | 0.72 | 24 | 1.20 (0.04) | 0.70 | 2.07 |
| | 2014 | 80 | 4 | 0.20 (0.02) | 0.08 | 0.53 | 25 | 1.25 (0.04) | 0.75 | 2.09 |
| | Total | 762 | 90 | 0.47 (0.002) | 0.36 | 0.62 | 276 | 1.44 (0.004) | 1.22 | 1.73 |
| Frost Ford RKM 291.8 RM 181.3 | 2004 | 60 | 3 | 0.20 (0.02) | 0.06 | 0.62 | 112 | 7.47 (0.10) | 6.07 | 9.19 |
| | 2005 | 60 | 5 | 0.33 (0.02) | 0.14 | 0.79 | 81 | 5.40 (0.10) | 4.09 | 7.12 |
| | 2006 | 60 | 5 | 0.33 (0.02) | 0.14 | 0.79 | 111 | 7.40 (0.14) | 5.55 | 9.86 |
| | 2007 | 60 | 9 | 0.60 (0.02) | 0.32 | 1.12 | 329 | 21.93 (0.33) | 17.38 | 27.68 |
| | 2008 | 56 | 12 | 0.86 (0.04) | 0.44 | 1.66 | 562 | 40.14 (0.48) | 33.62 | 47.93 |
| | 2009 | 91 | 17 | 0.75 (0.02) | 0.44 | 1.26 | 427 | 18.77 (0.17) | 15.77 | 22.34 |
| | 2010 | 60 | 9 | 0.60 (0.03) | 0.26 | 1.37 | 319 | 21.27 (0.28) | 17.39 | 26.00 |
| | 2011 | 82 | 19 | 0.93 (0.02) | 0.59 | 1.46 | 466 | 22.73 (0.23) | 18.89 | 27.35 |
| | 2012 | 80 | 15 | 0.75 (0.02) | 0.42 | 1.33 | 390 | 19.50 (0.20) | 16.31 | 23.31 |
| | 2013 | 80 | 9 | 0.45 (0.02) | 0.22 | 0.91 | 311 | 15.55 (0.15) | 13.04 | 18.54 |
| | 2014 | 80 | 13 | 0.65 (0.02) | 0.39 | 1.08 | 277 | 13.85 (0.13) | 11.77 | 16.30 |
| | Total | 769 | 116 | 0.60 (0.002) | 0.49 | 0.75 | 3,385 | 17.61 (0.02) | 16.26 | 19.07 |
| Wallen Bend RKM 309.6 RM 192.4 | 2004 | 60 | 3 | 0.20 (0.02) | 0.06 | 0.62 | 28 | 1.87 (0.05) | 1.27 | 2.75 |
| | 2005 | 60 | 3 | 0.20 (0.02) | 0.06 | 0.62 | 21 | 1.40 (0.04) | 0.86 | 2.27 |
| | 2006 | 60 | 1 | 0.07 (0.01) | 0.01 | 0.49 | 20 | 1.33 (0.04) | 0.82 | 2.17 |
| | 2007 | 60 | 5 | 0.33 (0.02) | 0.14 | 0.79 | 76 | 5.07 (0.09) | 3.90 | 6.58 |
| | 2008 | 60 | 10 | 0.67 (0.03) | 0.33 | 1.35 | 134 | 8.93 (0.15) | 6.84 | 11.67 |
| | 2009 | 60 | 8 | 0.53 (0.03) | 0.25 | 1.13 | 87 | 5.80 (0.10) | 4.42 | 7.62 |
| | 2010 | 60 | 4 | 0.27 (0.02) | 0.10 | 0.71 | 59 | 3.93 (0.08) | 2.92 | 5.31 |
| | 2011 | 88 | 5 | 0.23 (0.01) | 0.09 | 0.58 | 48 | 2.18 (0.04) | 1.57 | 3.03 |
| | 2012 | 81 | 5 | 0.25 (0.01) | 0.10 | 0.59 | 77 | 3.80 (0.08) | 2.66 | 5.44 |
| | 2013 | 84 | 1 | 0.05 (0.01) | 0.01 | 0.35 | 66 | 3.14 (0.06) | 2.17 | 4.55 |
| | 2014 | 84 | 5 | 0.24 (0.02) | 0.08 | 0.67 | 64 | 3.00 (0.07) | 1.96 | 4.58 |
| | Total | 757 | 50 | 0.26 (0.001) | 0.19 | 0.36 | 680 | 3.59 (0.01) | 3.17 | 4.07 |

*Data from 2004–2009 are in [3,4].

length anterior-posterior (nearest 0.1 mm) using precision calipers, and returned to their approximate position of collection.

Population density ($\hat{D}$), i.e., number of mussels per m², was calculated from the mean number of individuals of each species in quadrat samples excavated at each site [14–16]. Total area

**Table 2. Estimates of site-specific and total (pooled) population sizes for *Epioblasma brevidens* and *E. capsaeformis* for each annual census conducted in the Clinch River, TN from 2004–2014.** Population size ($\hat{N}$) per site was obtained by multiplying mussel density ($\hat{D}$) values in Table 1 by the site square area measured in square meters ($m^2$). RKM = River Kilometer, RM = River Mile.

| Site (Name, RKM) | Site Area (m²) | Census Year | *Epioblasma brevidens* | | *Epioblasma capsaeformis* | |
|---|---|---|---|---|---|---|
| | | | $\hat{N}$ | 95% CL (Lower–Upper) | $\hat{N}$ | 95% CL (Lower–Upper) |
| Swan Island | 5,760 | 2004 | 2,304 | (1,056–5,029) | 3,840 | (2,009–7,340) |
| RKM 277.1 | | 2005 | 768 | (189–3,118) | 3,456 | (1,724–6,929) |
| RM 172.2 | | 2006 | 1,152 | (371–3,581) | 768 | (189–3,118) |
| | | 2007 | 3,456 | (1,724–6,929) | 7,296 | (4,596–11,582) |
| | | 2008 | 1,536 | (460–5,132) | 4,224 | (2,183–8,174) |
| | | 2009 | 4,160 | (2,064–8,384) | 8,640 | (5,573–13,395) |
| | | 2010 | 2,304 | (1,056–5,029) | 10,752 | (7,114–16,251) |
| | | 2011 | 5,120 | (2,779–9,433) | 13,056 | (9,045–18,845) |
| | | 2012 | 4,608 | (2,413–8,799) | 20,160 | (14,436–28,153) |
| | | 2013 | 2,016 | (980–4,149) | 6,912 | (4,008–11,920) |
| | | 2014 | 1,152 | (435–3,052) | 7,200 | (4,310–12,028) |
| | | Mean | 2,598 | (1,230–5,694) | 7,846 | (5,017–12,521) |
| Frost Ford | 15,050 | 2004 | 3,010 | (968–9,364) | 112,373 | (91,305–138,304) |
| RKM 291.8 | | 2005 | 5,017 | (2,115–11,898) | 81,270 | (61,615–107,196) |
| RM 181.3 | | 2006 | 5,017 | (2,115–11,898) | 111,370 | (83,577–148,406) |
| | | 2007 | 9,030 | (4,859–16,783) | 330,097 | (261,606–416,520) |
| | | 2008 | 12,040 | (5,928–24,452) | 604,150 | (506,046–721,273) |
| | | 2009 | 11,246 | (6,667–18,971) | 282,477 | (237,298–336,258) |
| | | 2010 | 9,030 | (3,946–20,666) | 320,063 | (261,776–391,329) |
| | | 2011 | 13,949 | (8,857–21,967) | 342,112 | (284,340–411,622) |
| | | 2012 | 11,288 | (6,390–19,939) | 293,475 | (245,523–350,792) |
| | | 2013 | 6,773 | (3,354–13,675) | 234,028 | (196,262–279,060) |
| | | 2014 | 9,783 | (5,886–16,257) | 208,443 | (177,123–245,300) |
| Frost Ford cont. | | Mean | 8,744 | (4,644–16,897) | 265,442 | (218,770–322,369) |
| Wallen Bend | 3,182 | 2004 | 636 | (205–1,976) | 5,940 | (4,035–8,744) |
| RKM 309.6 | | 2005 | 636 | (205–1,976) | 4,455 | (2,741–7,239) |
| RM 192.4 | | 2006 | 212 | (29–1,562) | 4,243 | (2,609–6,899) |
| | | 2007 | 1,061 | (448–2,512) | 16,122 | (12,408–20,948) |
| | | 2008 | 2,121 | (1,046–4,302) | 28,426 | (21,757–37,139) |
| | | 2009 | 1,697 | (800–3,600) | 18,456 | (14,051–24,241) |
| | | 2010 | 849 | (321–2,244) | 12,516 | (9,278–16,884) |
| | | 2011 | 723 | (284–1,842) | 6,943 | (4,997–9,647) |
| | | 2012 | 786 | (331–1,865) | 12,100 | (8,450–17,325) |
| | | 2013 | 152 | (21–1,100) | 10,001 | (6,906–14,482) |
| | | 2014 | 796 | (298–2,127) | 9,546 | (6,248–14,584) |
| | | Mean | 879 | (362–2,282) | 11,704 | (8,498–16,194) |
| Total (Pooled) | 23,992 | 2004 | 5,950 | (2,229–16,369) | 122,153 | (97,349–154,388) |
| | | 2005 | 6,421 | (2,509–16,992) | 89,181 | (66,079–121,364) |
| | | 2006 | 6,381 | (2,515–17,041) | 116,381 | (86,375–158,423) |
| | | 2007 | 13,547 | (7,030–26,223) | 353,515 | (278,610–449,050) |
| | | 2008 | 15,697 | (7,434–33,886) | 636,800 | (529,986–766,586) |
| | | 2009 | 17,103 | (9,531–30,955) | 309,573 | (256,922–373,894) |
| | | 2010 | 12,183 | (5,323–27,939) | 343,331 | (278,168–424,464) |
| | | 2011 | 19,792 | (11,920–33,242) | 362,111 | (298,382–440,114) |

*(Continued)*

**Table 2.** (Continued)

| Site (Name, RKM) | Site Area (m²) | Census Year | Epioblasma brevidens | | Epioblasma capsaeformis | |
|---|---|---|---|---|---|---|
| | | | $\hat{N}$ | 95% CL (Lower–Upper) | $\hat{N}$ | 95% CL (Lower–Upper) |
| | | 2012 | 16,681 | (9,134–30,603) | 325,735 | (268,409–396,270) |
| | | 2013 | 8,940 | (4,355–18,924) | 250,941 | (207,176–305,462) |
| | | 2014 | 11,730 | (6,619–21,436) | 225,189 | (187,681–271,912) |
| | | Mean | 12,220 | (6,236–24,873) | 284,992 | (232,285–351,084) |

and density were used to estimate population size ($\hat{N}$) for each species per site. Each independent census was first tested for normality based on goodness-of-fit, using a Shapiro-Wilk test ($\alpha = 0.05$) in JMP 11.0 statistical software (SAS, Raleigh, North Carolina). Because the quadrat sample data was right skewed due to being inflated with zeroes and ones, a logarithmic transformation of $\hat{D}$ and a delta-method approximation of the variance was used to calculate 95% confidence limits (CL) using the following formula [16]:

$$\exp\left(\ln(\hat{N}) \pm t_{\frac{z}{2},df}\sqrt{\frac{\hat{var}(\hat{N})}{\hat{N}^2}}\right)$$

Site-specific and cumulative size-class composition (by sex and sexes combined), total population size, and sex-ratios of E. brevidens and E. capsaeformis were estimated across each of the 11 annual censuses. Discrete estimates of $\hat{N}$ and mean lengths were used to predict age and age-class specific demographic rates for both species. Total or grand means of $\hat{N}$ and total variances [Var ($\hat{N}$)] were calculated for all sites and for all three populations pooled for the entire study period. Sampling error variance [$S^2(\hat{N})$] was determined from the variance of the 11 annual population size estimates. True temporal (process) variation was then determined by calculating coefficients of variation, according to the square-root of the total variance values minus $S^2(\hat{N})$ and dividing by the global mean values [17,18]. A generalized linear model (GLM) was used to test for significance of trends in the time series data collected from 2004–2014 using JMP 11.0 statistical software. The model was implemented using a Poisson distribution and the log-link function. The Pearson correlation coefficient was used to test for positive or negative correlation between random samples.

Ages of live mussels were estimated using von Bertallanfy growth equations [19] specific to females and males of each species; the equations and associated shell growth parameters were derived from shell thin-sections to determine age [4]. Trends in mean shell length observed for all individuals were compared over the study period. Population growth rates were calculated for both adults (Ages ≥4 yrs) and juveniles (Ages 1–3 yrs) combined and separately for adults only for each interval between annual censuses. Then, geometric mean growth rate spanning the entire 11-year period was calculated (Tables 3 and 4), using standard demographic procedures and the following three equations:

$$\lambda = \hat{N}_{t+1}/\hat{N}_t \tag{1}$$

$$r = \ln(\lambda) \tag{2}$$

$$\lambda_G = e\bar{r} \tag{3}$$

**Table 3. Estimated population size per census ($\hat{N}_t$), finite rate of population increase [$\lambda_{t-t+1}$] per time step, per capita or instantaneous rate of increase [$\bar{r}$], and cumulative population growth observed in populations of Epioblasma brevidens in the Clinch River, TN at Swan Island, Frost Ford, Wallen Bend, and Total (Sites Pooled) from 2004–2014.**

| Census (t) | Swan Island | | | | Frost Ford | | | | Wallen Bend | | | | Total (Sites Pooled) | | | |
|---|---|---|---|---|---|---|---|---|---|---|---|---|---|---|---|---|
| | All Individuals | | Adults Only | | All Individuals | | Adults Only | | All Individuals | | Adults Only | | All Individuals | | Adults Only | |
| | ($\hat{N}_t$) | $\lambda_{t-t+1}$ | ($\hat{N}_t$) | $\lambda_{t-t+1}$ | ($\hat{N}_t$) | $\lambda_{t-t+1}$ | ($\hat{N}_t$) | $\lambda_{t-t+1}$ | ($\hat{N}_t$) | $\lambda_{t-t+1}$ | ($\hat{N}_t$) | $\lambda_{t-t+1}$ | ($\hat{N}_t$) | $\lambda_{t-t+1}$ | ($\hat{N}_t$) | $\lambda_{t-t+1}$ |
| 2004 (0) | 2,304 | N/A | 2,304 | N/A | 3,010 | N/A | 1,003 | N/A | 636 | N/A | 212 | N/A | 5,950 | N/A | 3,519 | N/A |
| 2005 (1) | 768 | 0.33 | 384 | 0.17 | 5,017 | 1.67 | 4,014 | 4.00 | 636 | 1.00 | 636 | 3.00 | 6,421 | 1.08 | 5,034 | 1.43 |
| 2006 (2) | 1,152 | 1.50 | 384 | 1.00 | 5,017 | 1.00 | 3,010 | 0.75 | 212 | 0.33 | 212 | 0.33 | 6,381 | 0.99 | 3,606 | 0.72 |
| 2007 (3) | 3,456 | 3.00 | 1,536 | 4.00 | 9,030 | 1.80 | 7,023 | 2.33 | 1,061 | 5.00 | 212 | 1.00 | 13,547 | 2.12 | 8,772 | 2.43 |
| 2008 (4) | 1,536 | 0.44 | 768 | 0.50 | 12,040 | 1.33 | 7,023 | 1.00 | 2,121 | 2.00 | 636 | 3.00 | 17,606 | 1.16 | 8,428 | 0.96 |
| 2009 (5) | 4,160 | 2.71 | 3,520 | 4.58 | 11,246 | 0.87 | 7,277 | 1.04 | 1,697 | 0.80 | 636 | 1.00 | 17,103 | 1.09 | 11,433 | 1.36 |
| 2010 (6) | 2,304 | 0.55 | 2,304 | 0.65 | 9,030 | 0.80 | 6,020 | 0.83 | 849 | 0.50 | 637 | 1.00 | 12,183 | 0.71 | 8,961 | 0.78 |
| 2011 (7) | 5,120 | 2.22 | 4,352 | 1.89 | 13,949 | 1.54 | 11,012 | 1.83 | 723 | 0.85 | 578 | 0.91 | 19,792 | 1.62 | 15,943 | 1.78 |
| 2012 (8) | 4,608 | 0.90 | 3,744 | 0.86 | 11,288 | 0.81 | 9,783 | 0.89 | 786 | 1.09 | 786 | 1.36 | 16,681 | 0.84 | 14,313 | 0.90 |
| 2013 (9) | 2,016 | 0.44 | 1,728 | 0.46 | 6,773 | 0.60 | 6,773 | 0.69 | 152 | 0.19 | 152 | 0.19 | 8,940 | 0.54 | 8,653 | 0.60 |
| 2014 (10) | 1,152 | 0.57 | 1,152 | 0.67 | 9,783 | 1.44 | 9,030 | 1.33 | 758 | 5.00 | 796 | 5.24 | 11,730 | 1.31 | 10,978 | 1.27 |
| Instantaneous/Annual Growth | | | | | | | | | | | | | | | | |
| $\bar{r}(\hat{\sigma}^2)$ | -0.07 (0.26) | | -0.07 (0.32) | | 0.12 (0.11) | | 0.22 (0.18) | | 0.02 (0.34) | | 0.13 (0.32) | | 0.07 (0.12) | | 0.11 (0.14) | |
| $\lambda_G$ | 0.93 | | 0.93 | | 1.13 | | 1.25 | | 1.02 | | 1.14 | | 1.07 | | 1.12 | |
| %Growth$_{Annual}$ | -6.7% | | -6.7% | | 13% | | 24.6% | | 2% | | 14.1% | | 7% | | 12.0% | |
| Cumulative Growth from 2004–2014 | | | | | | | | | | | | | | | | |
| $r_{2004-14}$ | -0.69 | | -0.69 | | 1.18 | | 2.20 | | 0.17 | | 1.32 | | 0.68 | | 1.14 | |
| $\lambda_{2004-14}$ | 0.50 | | 0.50 | | 3.25 | | 9.00 | | 1.19 | | 3.75 | | 1.97 | | 3.12 | |
| %Growth$_{2004-14}$ | -50% | | -50% | | 225% | | 800% | | 19% | | 275% | | 97% | | 212% | |

where λ is the finite annual population growth rate (Eq 1), calculated using $\hat{N}_t$ as the number of individuals (or density) in the population at year $t$, and $\hat{N}_{t+1}$ as the number of individuals in the next census year [18,20]. The natural log (ln) of λ (Eq 2), more commonly referred to as the instantaneous annual population growth rate or $r$, was used to compute the arithmetic mean $\bar{r}$, and the associated variance ($\hat{\sigma}^2$), and then these respective values were transformed back to obtain the mean finite annual population growth rate ($\lambda_G$) and the mean associated variance measures using inverse $\log_e$ (Eq 3). The Durbin-Watson statistic $d$ was used to test for strength of temporal autocorrelation in the data.

To estimate mortality, a catch-curve regression analysis was conducted based on number-at-age to compute annual mortality for each species:

$$ln(\hat{N}_t) = \ln(\hat{N}_0) - Zt$$

where $\hat{N}_t$ is the total number (i.e., frequency) in an age class at time $t$ obtained from all three sites and 11 census (see Fig 4), and similarly $\hat{N}_0$ is the original number in an age class, and $Z$ is the annual instantaneous mortality rate [21]. This procedure is a simple linear regression [$y = a—bx$], where the slope ($b$) is equivalent to $Z$. The annual instantaneous mortality rate ($Z$) was converted to the annual finite mortality rate ($A$), where $A = 1—e^{-Z}$. The catch-curve analysis is analogous to the life-table analysis used by ecologists. To ensure a larger sample size encompassing a range of age and size classes and to obtain a more precise estimate of mortality, estimates were obtained with both sexes combined. In addition, sex-specific estimates also were calculated to investigate mortality of males and females of each species. Assumptions of the

**Table 4. Estimated population size per census ($\hat{N}_t$), finite rate of population increase [$\lambda_{t-t+1}$] per time step, per capita or instantaneous rate of increase [$\bar{r}$], and cumulative population growth observed in populations of Epioblasma capsaeformis in the Clinch River, TN at Swan Island, Frost Ford, Wallen Bend, and Total (Sites Pooled) from 2004–2014.**

| Census (t) | Swan Island | | | | Frost Ford | | | | Wallen Bend | | | | Total (Sites Pooled) | | | |
|---|---|---|---|---|---|---|---|---|---|---|---|---|---|---|---|---|
| | All Individuals | | Adults Only | | All Individuals | | Adults Only | | All Individuals | | Adults Only | | All Individuals | | Adults Only | |
| | ($\hat{N}_t$) | $\lambda_{t-t+1}$ | ($\hat{N}_t$) | $\lambda_{t-t+1}$ | ($\hat{N}_t$) | $\lambda_{t-t+1}$ | ($\hat{N}_t$) | $\lambda_{t-t+1}$ | ($\hat{N}_t$) | $\lambda_{t-t+1}$ | ($\hat{N}_t$) | $\lambda_{t-t+1}$ | ($\hat{N}_t$) | $\lambda_{t-t+1}$ | ($\hat{N}_t$) | $\lambda_{t-t+1}$ |
| 2004 (0) | 3,840 | N/A | 3,840 | N/A | 112,373 | N/A | 108,360 | N/A | 5,940 | N/A | 5,940 | N/A | 122,153 | N/A | 118,140 | N/A |
| 2005 (1) | 3,456 | 0.90 | 2,688 | 0.70 | 81,270 | 0.72 | 72,240 | 0.67 | 4,455 | 0.75 | 3,606 | 0.61 | 89,181 | 0.73 | 78,534 | 0.66 |
| 2006 (2) | 768 | 0.22 | 384 | 0.14 | 111,370 | 1.37 | 79,263 | 1.10 | 4,243 | 0.95 | 3,182 | 0.88 | 116,381 | 1.30 | 82,830 | 1.05 |
| 2007 (3) | 7,296 | 9.50 | 3,840 | 10.00 | 330,097 | 2.96 | 141,470 | 1.78 | 16,122 | 3.80 | 4,879 | 1.53 | 353,515 | 3.04 | 150,189 | 1.81 |
| 2008 (4) | 4,224 | 0.58 | 2,688 | 0.70 | 604,150 | 1.83 | 190,275 | 1.34 | 28,426 | 1.76 | 8,485 | 1.74 | 636,800 | 1.80 | 201,448 | 1.34 |
| 2009 (5) | 8,640 | 2.05 | 6,720 | 2.50 | 282,477 | 0.47 | 110,477 | 0.58 | 18,456 | 0.65 | 9,970 | 1.18 | 309,573 | 0.49 | 127,167 | 0.63 |
| 2010 (6) | 10,752 | 1.24 | 7,296 | 1.09 | 320,063 | 1.13 | 232,773 | 2.11 | 12,516 | 0.68 | 10,395 | 1.04 | 343,331 | 1.11 | 250,464 | 1.97 |
| 2011 (7) | 13,056 | 1.21 | 7,168 | 0.98 | 342,112 | 1.07 | 299,532 | 1.29 | 6,943 | 0.55 | 4,629 | 0.45 | 362,111 | 1.05 | 311,328 | 1.24 |
| 2012 (8) | 20,160 | 1.54 | 6,912 | 0.96 | 293,475 | 0.86 | 228,008 | 0.76 | 12,099 | 1.74 | 6,128 | 1.32 | 325,734 | 0.90 | 241,048 | 0.77 |
| 2013 (9) | 6,912 | 0.34 | 4,032 | 0.58 | 234,028 | 0.80 | 174,580 | 0.77 | 10,001 | 0.83 | 5,910 | 0.96 | 250,940 | 0.77 | 184,522 | 0.77 |
| 2014 (10) | 7,200 | 1.04 | 6,624 | 1.64 | 208,443 | 0.89 | 156,520 | 0.90 | 9,546 | 0.95 | 6,265 | 1.06 | 225,189 | 0.90 | 169,409 | 0.92 |
| Instantaneous/Annual Growth | | | | | | | | | | | | | | | | |
| $\bar{r}(\hat{\sigma}^2)$ | 0.06 (0.16) | | 0.05 (0.34) | | 0.06 (0.32) | | 0.04 (0.14) | | 0.05 (0.19) | | 0.01 (0.13) | | 0.06 (0.16) | | 0.04 (0.13) | |
| $\lambda_G$ | 1.06 | | 1.06 | | 1.06 | | 1.04 | | 1.05 | | 1.01 | | 1.06 | | 1.04 | |
| %Growth$_{Annual}$ | 6.5% | | 5.6% | | 6.4% | | 3.7% | | 4.9% | | 0.5% | | 6.3% | | 3.7% | |
| Cumulative Growth from 2004–2014 | | | | | | | | | | | | | | | | |
| $r_{2004-14}$ | 0.63 | | 0.55 | | 0.62 | | 0.37 | | 0.47 | | 0.05 | | 0.61 | | 0.36 | |
| $\lambda_{2004-14}$ | 1.88 | | 1.73 | | 1.85 | | 1.44 | | 1.61 | | 1.05 | | 1.84 | | 1.43 | |
| %Growth$_{2004-14}$ | 88% | | 73% | | 85% | | 44% | | 61% | | 5% | | 84% | | 43% | |

catch-curve regression to estimate mortality are (1) constant recruitment; (2) equal survival among year classes; (3) constant natural mortality each year and among all year classes; and (4) catch-curves are fitted to samples representative of the true age structure of the population [21]. Age-0 mussels typically <10 mm long were not used to estimate mortality. As immature age classes (0–3 yrs) can violate one or more of these assumptions, analyses were conducted two ways, first using age classes ≥1 yrs to estimate mortality of juveniles plus the adults and then using age classes ≥4 yrs to estimate mortality of adults only.

Recruitment was defined in this study as the percentage of Age-1 individuals (>15 mm) relative to the total estimated number of individuals per site. Age-0 individuals typically were too small (<10 mm) to be reliably sampled, so they were excluded from this analysis. Recruitment

**Table 5. Estimated fecundity of female *Epioblasma brevidens* and *E. capsaeformis* sampled from the Clinch River, TN in May–2013 and female *E. capsaeformis* sampled from the Clinch River, TN, in May 2002 and May 2013.** Data from 2002 are in [10]. Fecundity was estimated by counting the number of glochidia in gravid female (♀) mussels. SE = Standard error of glochidia per ♀.

| Mussel species (sample year) | No. ♀ sampled (N) | Mean ♀ shell length (mm) | Range of ♀ shell length (mm) | Mean no. glochidia/♀ | SE | 95% CI | Range in no. glochidia/♀ |
|---|---|---|---|---|---|---|---|
| *E. brevidens* (2013) | 15 | 46.4 | 42.1–52.2 | 34,947 | 2,492 | ±4,884 | 18,987–56,151 |
| *E. capsaeformis* (2002) | 10 | 41.5 | 36.7–46.4 | 13,008* | 1,460 | ±2,862 | 7,780–16,876 |
| *E. capsaeformis* (2013) | 20 | 39.7 | 35.0–46.1 | 9,558* | 603 | ±1,182 | 3,456–22,182 |
| *E. capsaeformis* (Samples Pooled) | 30 | 40.3 | 35.0–46.4 | 10,708 | 785 | ±1,539 | 3,456–22,182 |

*Mean fecundity for *E. capsaeformis* sampled in 2002 and 2013 were not significantly different ($p = 0.1238$; mean values are plotted and compared in S6 File).

was calculated by using data from each of the respective sites and dividing number of Age-1 individuals in a census by total number of individual's ≥Age-2 in the previous census. The approximate smallest (lower bound) observed size of Age-1 individuals, which was set at ≥15 mm, and the predicted age-at-length boundary between Age-1 and Age-2 individuals (upper bound of Age-1 and lower bound of Age-2, which was set at ≤25 mm) was used to categorize Age-1 individuals for each species [2,4].

## Female fecundity

A fourth Hancock County, TN, site on the Clinch River, Kyles Ford (KF), RKM 305.1 (RM 189.6), was used to assess fecundity of gravid females of each species (Table 5; Fig 1). Glochidia were extracted on 30 May 2013, when glochidial brooding of females was considered to be at its apex for each species. The site was selected because the population size of these mussels is large and thus an adequate sample size of each species representing multiple age-classes could be obtained. Glochidia were removed from one gill of each individual collected. Total fecundity was then estimated by multiplying the glochidia sample count by two to determine total fecundity. Glochidia from one fully gravid gill were extracted from each of 20 *E. capsaeformis* and 15 *E. brevidens*. Glochidia extraction was conducted by using a hypodermic syringe and needle to inject river water into the gill, beginning anteriorly and slowly moving posteriorly until all glochidia were removed from the gill. Mussels were then rinsed out with river water to ensure no glochidia were trapped inside the body. The larvae from each mussel were flushed directly into a clean petri dish. After pouring away the excess river water, the contents of the petri dish were carefully poured into glass vials and preserved in 95% ethanol. Sample vials were labeled with each female's species name and maximum shell length (mm), and transported to the Freshwater Mollusk Conservation Center, Blacksburg, VA, for analysis. Samples were transferred to a gridded petri dish and dispersed uniformly prior to counting under an Olympus® SZ40 microscope. Three complete counts of each sample were made using hand-held manual mechanical counters while systematically covering each grid square. An average of the three counts was taken as the final count of each gill sample. Age was estimated for each female based on sex-specific von Bertalanffy growth curves [4]. Regression analysis was conducted to evaluate the linear relationship of total length (mm) and age (yrs) to fecundity in all females sampled. An additional 10 samples of *E. capsaeformis* were acquired from [4] and included in the analysis to increase statistical power (Table 5). Following analysis of the linear relationship of length-to-fecundity, the proportions of females in each age class were used to extrapolate fecundity estimates to the total number of glochidia brooded by all adult females at monitoring sites each census year.

## Clinch River discharge data

Stream discharge was compared to trends in mussel population growth over the 11 census years. Daily mean discharge—measured in cubic feet per second (ft$^3$/s or more commonly, cfs) —was obtained from United States Geological Survey (USGS) Gage #03528000 in Tazewell, TN. Data available from 1 October 1930–30 September 2014 was downloaded at: http://waterdata.usgs.gov/nwis. This dataset was the most complete and nearest to the study area (the gage is located ~19 km downstream of SI). Daily mean discharge was binned into annual intervals from October 1 to September 30, which matched census intervals. Using the entire 84-year dataset (n = 30,681), several statistics were calculated to characterize annual discharge variation using the Indicators of Hydrologic Alteration (IHA) program [22], to include monthly median discharge, exceedance probabilities for the daily mean discharge, extreme low discharge (discharge defined as 10th percentile), low discharge (25th percentile), extreme

high discharge (90^{th} percentile), 90-day minimal discharge, and 90-day maximal discharge. Many statistics provided in IHA were not used because of high correlation among or lack of information provided for the study period. Additional discharge statistics were derived from IHA statistics to better described annual variation in the study period with respect to the entire dataset. They were number of days in a year with discharge classified as extremely low, low, extremely high, and below median according to the entire discharge dataset. Median annual discharge for every year was also calculated. The IHA program also was used to determine whether the study period (1 October 2004 to 20 September 2014) differed from the preceding 74 years.

Hydrographs, correlations among stream discharge statistics, as well as, correlations between discharge statistics and demographic parameters were examined and presented as a set of complete hypotheses. Specifically, Pearson Product-Moment Correlation (Pearson's r) statistics were analyzed using JMP 15.2.1 software (SAS Institute, Inc. Cary, NC). Preliminary inspection of hydrographs and correlations resulted in the calculation of an additional statistic, median discharge from June 1 to September 30. Since observed correlations were generally linear, models were fit for each hypothesis and compared to each other and to a null model (intercept only) using Akaike's Information Criterion corrected for sample size (AICc) in JMP 15.2.1. Models with greater relative support were graphed.

## Results

### Population size and density

Mean population size ($\hat{N}$) of *E. brevidens* at the three investigated sites from 2004–2014 was 12,295 (95% CL = 6,236–24,873) individuals, with moderate to large differences observed among sites (Table 2). The pooled estimates of population size from all sites ranged from 5,950 in 2004 to 17,103 in 2009 (Table 2; Fig 2). Per-site population size ranged from a minimum of 152 individuals at WB in 2013 to a maximum of 13,949 individuals at FF in 2011 with 95% CL greater than ±50% of the means. At each site, population sizes were stable from 2004–2014 and increased moderately over the study period; however, trend increases were not statistically significant at any site. During specific time intervals, decreases in population size were observed at all sites from 2005–2006, 2009–2010, and 2012–2013; these site-level declines were not significant (Fig 2). However, when combining $\hat{N}$ from all sites (pooled), significant increases ($p<0.001$) in population size were detected between 2006 and 2009 and a significant decrease ($p<0.001$) occurred between 2011 and 2013. Among sites and census years, density always remained $<1$ m$^{-2}$ (range = 0.13–0.93 m$^{-2}$) (Fig 2). Temporal autocorrelation was not detected in the time series censuses ($\hat{N}$) at sites individually or pooled.

Pooled mean population size ($\hat{N}$) of *E. capsaeformis* at the three investigated sites from 2004–2014 was 284,992 (95% CL = 232,285–351,083), with large differences observed within and among sites (Table 2). The pooled estimates of population size from all sites ranged from 89,181 in 2004 to 636,800 in 2009 (Table 2; Fig 3). Population size across sites ranged from a minimum of 768 individuals at SI in 2006 to a maximum of 604,150 at FF in 2008, with 95% CL typically ±50% of the means. At FF and WB, population size was stable from 2004–2006, and then increased significantly from 2006–2008 ($p<0.001$). From 2008–2009, population size at FF significantly decreased ($p<0.001$) but was stable from 2009–2014. At WB, population size decreased significantly ($p<0.001$) from 2008–2011, after which, the population increased slightly before stabilizing. At SI, population size of *E. capsaeformis* was stable from 2004–2006, and then increased significantly from 2006–2007 ($p<0.001$). Unlike the FF and WB populations, the SI population remained stable from 2007–2012 but slightly increased from 2008–

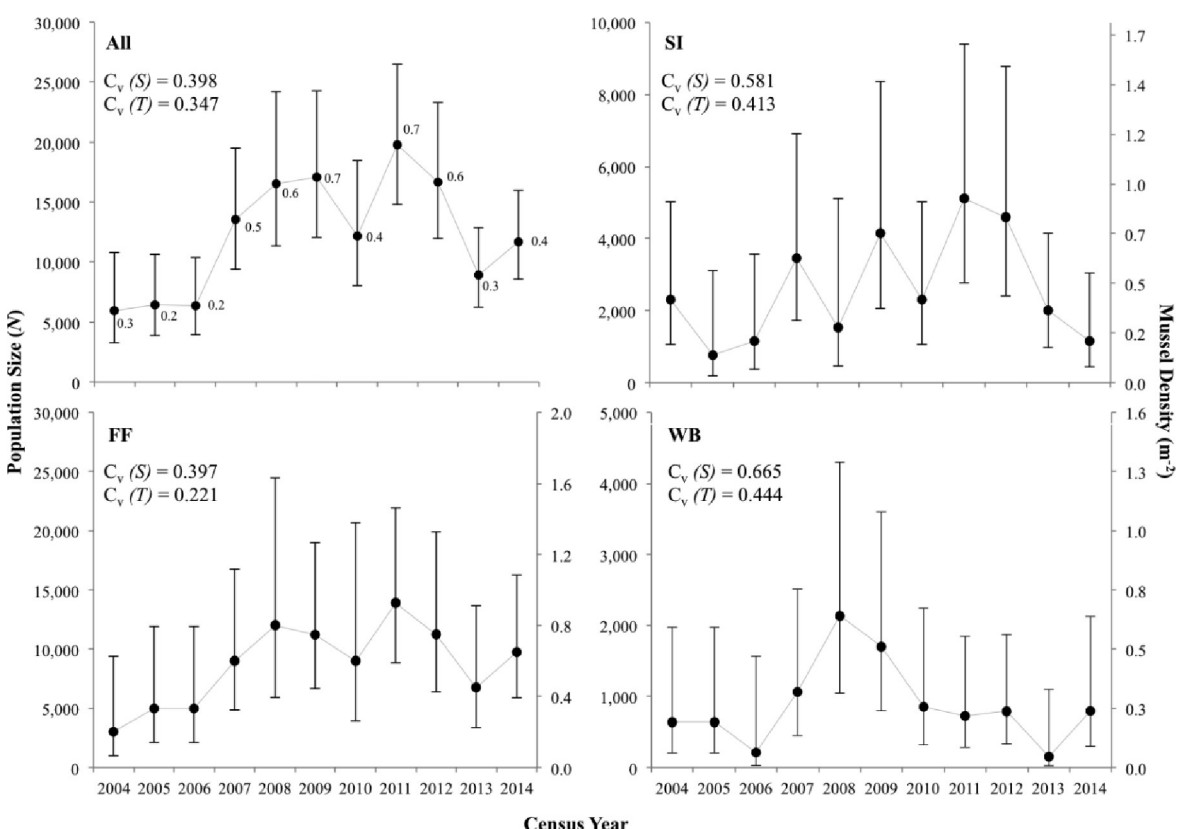

**Fig 2. Estimates of population size and mean density (bars represent 95% confidence limits corrected for non-normality) for** *Epioblasma brevidens* **in the Clinch River, Hancock County, TN, at Frost Ford (FF), Swan Island (SI), Wallen Bend (WB), and population size averaged across all three sites (All) from 2004–2014.** Numbers next to point estimates on All plot indicate average density across the 3 sites. The value $C_v$ *(S)* is the total coefficient of variation measured in the time series, which includes both sampling error variance $S^2$ $(\hat{N})$ and true temporal process variance. The value $C_v$ *(T)* is true temporal process variation, after mean $S^2$ $(\hat{N})$ was removed from $C_v$ *(S)*.

2012. From 2012–2013, the SI population decreased but not significantly, before stabilizing again from 2013–2014 (Fig 3). Correspondingly, density ranged widely from 0.1–3.5 m$^{-2}$ at SI, 5.4–40.1 m$^{-2}$ at FF, and 1.3–8.9 m$^{-2}$ at WB. Density typically exceeded 1 m$^{-2}$ at all sites over all time intervals, except from 2004–2006 at SI. When combining $\hat{N}$ from all sites (pooled), significant increases ($p<0.001$) in population size were detected from 2006–2008 and significant decreases ($p<0.001$) occurred from 2008–2013. Temporal autocorrelation was not detected in the census ($\hat{N}$) time series at SI, FF, or for the pooled population. However, two consecutive time intervals at WB did show significant temporal autocorrelation between density estimates, including 2006–2007 ($p = 0.0365$) and 2007–2008 ($p = 0.0270$). A third time interval at WB, from 2008–2009 was nearly significant ($p = 0.0509$).

## Population age structure and sex ratios

Mean age-class frequency of *E. brevidens* across all census years (2004–2014) and sites showed greater numbers of middle-aged individuals compared to young and older-aged individuals (Fig 4; S1 File). Mean age-class frequencies per year from 2004–2008 for *E. brevidens* were comprised of a higher frequency of young individuals, followed by a transition to higher frequencies of older individuals from 2008–2014, though sample sizes were small (e.g., 1

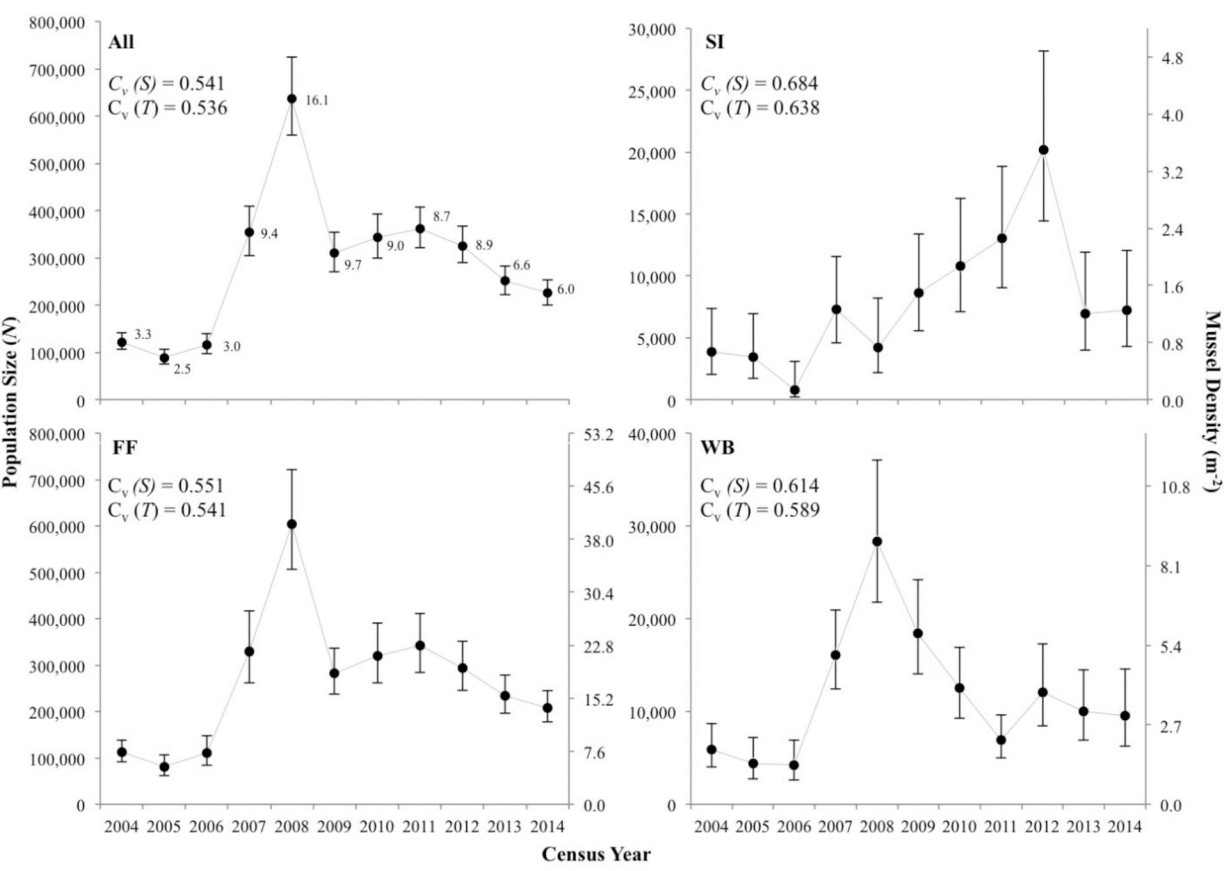

**Fig 3. Estimates of population size (*N*) and mean density (bars represent 95% confidence limits corrected for non-normality) for** *Epioblasma capsaeformis* **in the Clinch River, Hancock County, TN, at Frost Ford (FF), Swan Island (SI), Wallen Bend (WB), and averaged across all three sites (All) from 2004–2014.** Numbers next to point estimates on All plot indicate average density across the 3 sites. The value $C_v$ (S) is the total coefficient of variation measured in the time series, which includes both sampling error variance $S^2(\hat{N})$ and true temporal process variance. The value $C_v$ (T) is true temporal process variation, after mean $S^2(\hat{N})$ was removed from $C_v$ (S).

individual at WB in 2006 and again in 2013), making year-to-year age-class frequency comparisons for this species difficult (S1 File). Mean shell lengths for female *E. brevidens* was 42.2 mm (*N* = 89) and ranged from 16–58 mm, while for males it was 43.5 mm (*N* = 143) and ranged from 15–64 mm. Predicted ages ranged from 0–15 years for females and 0–20 years for males. Mean length-class frequencies of males and females are given separately and were larger for males (Fig 5; S2 File). From 2004–2014, mean sex ratios of *E. brevidens* was 40% females to 60% males. However, sex ratios varied greatly from 2004–2010, as the proportion of females ranged from a high of 67% in 2005 to a low of 26% in 2009. From 2012–2014, when sample sizes were increased at all sites, pooled sex ratios were approximately 40% females to 60% males, matching the pooled mean across all censuses and sites (Fig 7).

Mean age-class frequency of *E. capsaeformis* across all census years (2004–2014) and sites was skewed toward younger individuals (Fig 4; S1 File). Mean shell lengths for female *E. capsaeformis* was 34.2 mm (*N* = 1,485) and ranged from 11–46 mm, while for males it was 43.5 mm (*N* = 2,055) and ranged from 12–49 mm (Fig 5). This trend of younger individuals was especially evident from 2004–2008 in the density of juveniles versus adults (Fig 6), followed by a more evenly distributed age-class frequency from 2009–2011, and then finally shifting to a higher frequency of older individuals from 2012–2014 (S1 File). Predicted ages ranged from

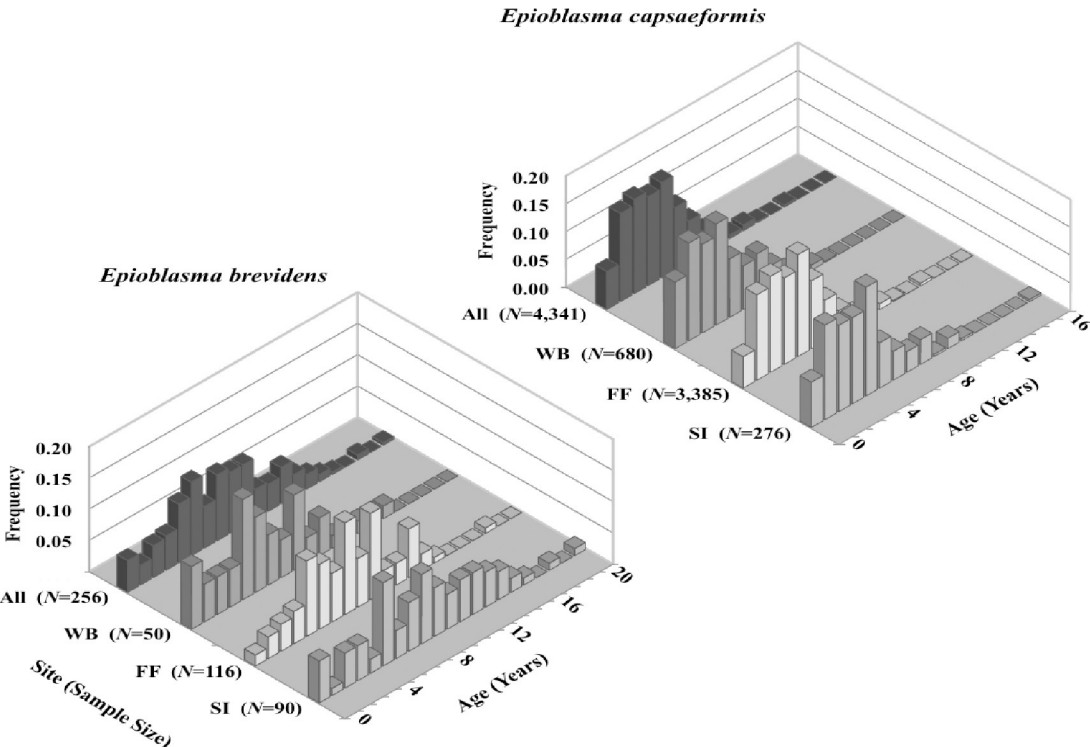

**Fig 4. Mean age class frequencies of *Epioblasma brevidens* and *E. capsaeformis* at Swan Island (SI), Frost Ford (FF), Wallen Bend (WB), and all sites pooled in the Clinch River, Hancock County, TN, from 2004–2014.** Ages were predicted from shell length (mm) using sex-specific von Bertalanffy growth curve equations in [4].

0–13 years for females and 0–16 years for males. Mean sex ratios for *E. capsaeformis* was similar to *E. brevidens*, and was comprised of 42% females to 58% males. However, in 2009 the sex ratio was 27% females to 73% males, which was highly atypical to the 10-year trend and coincided with a sharp decrease in densities at FF, WB, and the pooled population. When this datum was removed, the mean sex ratio was 45% females to 55% males and aligned with the mean estimates (Fig 7).

## Total and temporal/process variation observed across censuses

Total variation observed in *E. brevidens* population size ($\hat{N}$) per site and per year was strongly influenced by variance attributed to sampling error. Percent variance in sampling error relative to total variance in $\hat{N}$ for *E. brevidens* was 49% at SI, 69% at FF, 55% at WB, and 24% for the pooled $\hat{N}$ from 2004–2014. Subtracting mean sampling variance Var ($\hat{N}$) from total variance $S^2$ ($\hat{N}$) resulted in temporal coefficient of variations $Cv$ (T) for *E. brevidens* of 0.41 at SI, 0.22 at FF, 0.44 at WB, and 0.35 for all sites combined (Fig 2). Temporal variation from 2004–2008 [4] was higher than the period from 2009–2014, decreasing from 0.45 to 0.22 (S3 File).

Total variation observed in *E. capsaeformis* population size ($\hat{N}$) per site and per year was less influenced by variance from sampling error. Percent variance in sampling error relative to total variance in $\hat{N}$ for *E. capsaeformis* was 13% at SI, 4% at FF, 8% at WB, and 2% for the pooled $\hat{N}$ from 2004–2014. Subtracting mean sampling variance from total variance resulted in temporal coefficient of variations for *E. capsaeformis* of 0.64 at SI, 0.54 at FF, 0.59 at WB, and 0.54 across all sites (Fig 3). Similar to the trend observed in *E. brevidens*, temporal

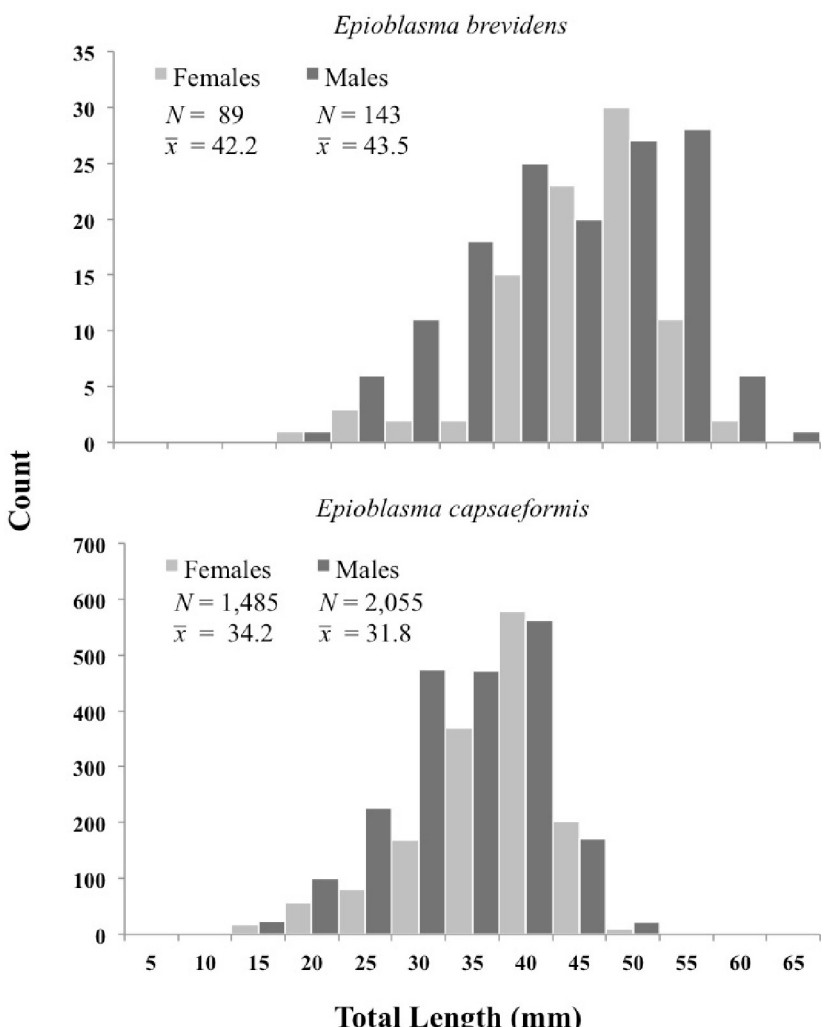

**Fig 5.** Mean ($\bar{x}$) length frequencies for the total sampled ($N$) males and females of *Epioblasma brevidens* and *E. capsaeformis* from all quadrat samples at Swan Island, Frost Ford, and Wallen Bend in the Clinch River, Hancock County, TN, from 2004–2014.

variation in $\hat{N}$ observed from 2004–2008 was higher than the period from 2009–2014, decreasing from 0.88 to 0.17 (S3 File).

## Population growth and mortality rates

From 2004–2014 for the three sites combined, mean annual population growth rate of *E. brevidens* was 7% and cumulative population growth was 97%. Mean annual population growth rate varied greatly among sites over the study period. For example, it was -6.7% at SI, 12.5% at FF, and 2% at WB and total population growth was -50% at SI, 225% at FF, and 19% at WB, respectively. Variance of the mean realized per-capita growth rates at SI ($\bar{r}$ = -0.07; $\hat{\sigma}^2$ = 0.26) and WB ($\bar{r}$ = .02; $\hat{\sigma}^2$ = 0.34) were more than twice their mean, while variance at FF ($\bar{r}$ = 0.12; $\hat{\sigma}^2$ = 0.11) and for the total combined (pooled) population ($\bar{r}$ = 0.07; $\hat{\sigma}^2$ = 0.12) were lower (Table 3). Recruitment of Age-1 *E. brevidens* ranged from 0.0% in 2013 and 2014 to 21.7% in 2007. Mean recruitment estimated across all years and sites was 9.1% (SE = 1.2). During this period, recruitment was variable, while population size of adults remained stable (Table 3; Fig

**(A)** *Epioblasma brevidens*

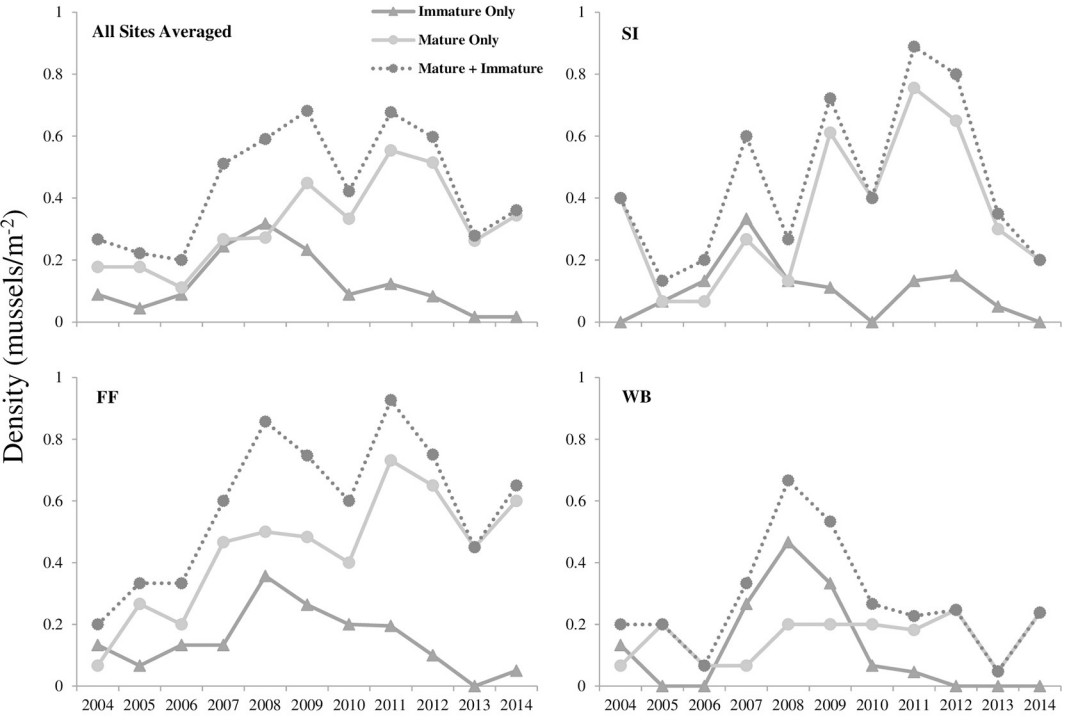

Census Year

**(B)** *Epioblasma capsaeformis*

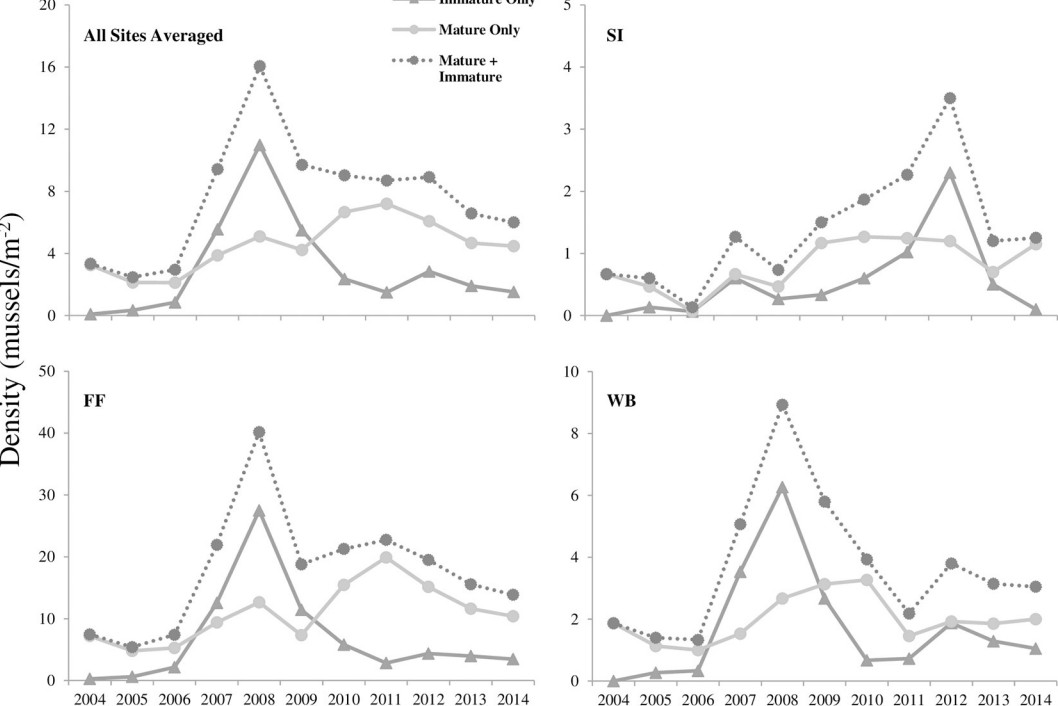

Census Year

**Fig 6.** Densities of (A) *Epioblasma brevidens* and (B) *Epioblasma capsaeformis* showing the contributions of immature recruits (<2 years old) versus only mature adults to fluctuations in population density at Swan Island (SI), Frost Ford (FF), Wallen Bend (WB), and all combined (pooled average) in the Clinch River, Hancock County, Tennessee from 2004–2014. For *Epioblasma brevidens* no recruits were observed at Wallen Bend in census years 2005–2006 or 2012–2014.

6; S1 File). There was no significant correlation between number of adults and number of recruits sampled in subsequent censuses. For ages 1–20, mean annual mortality for *E. brevidens* was 13.6% (SE = 3.2%) based on catch-curve regression analysis. However, when immature age classes (1–3 years) were removed from the analysis, annual mortality increased to 20.2% (SE = 2.8%) (Fig 8).

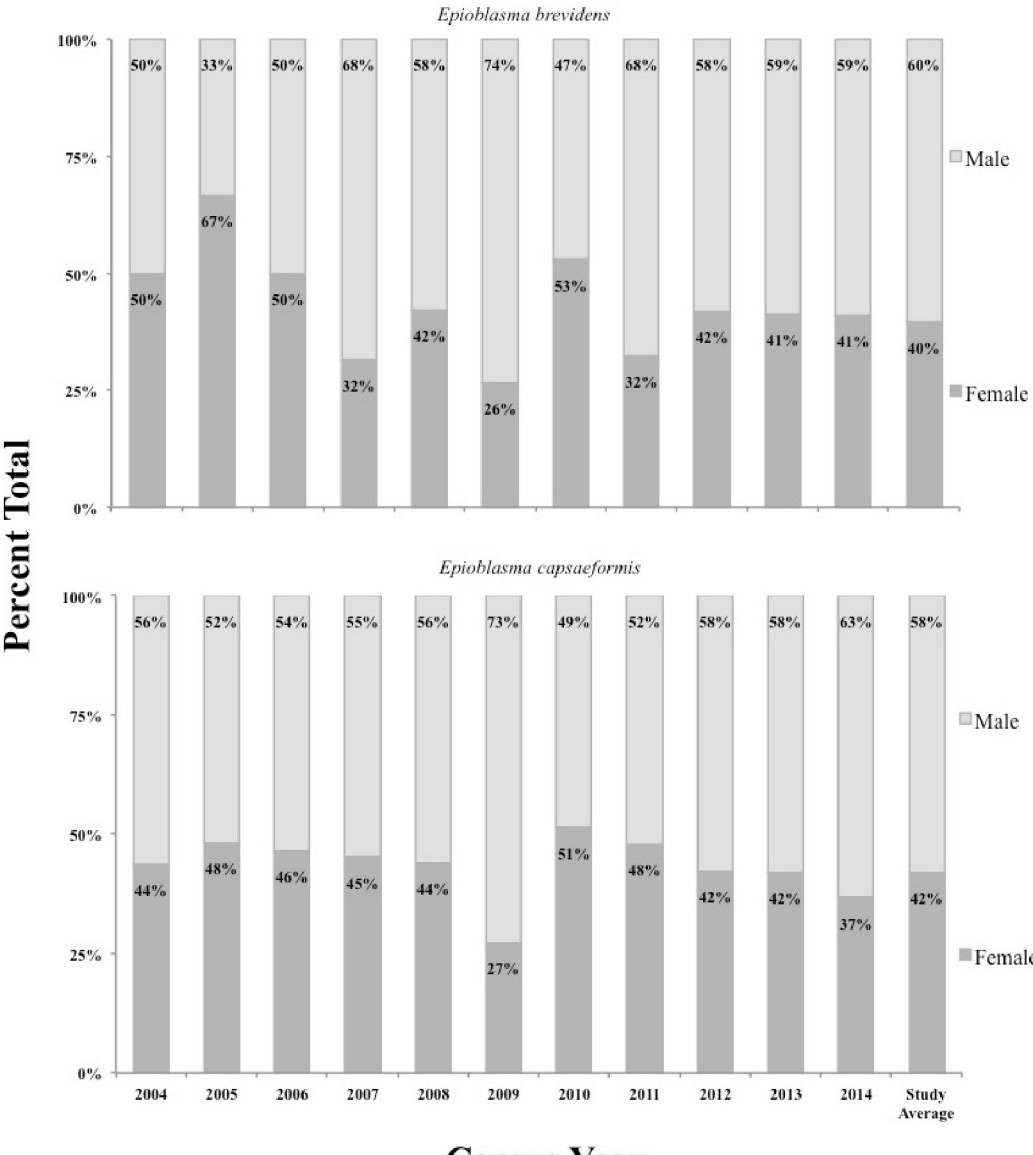

**Fig 7. Sex-ratios observed from all quadrat samples of *Epioblasma brevidens* and *E. capsaeformis* at Swan Island, Frost Ford, and Wallen Bend in the Clinch River, TN, from 2004–2014.** Calculations do not include individuals too small to differentiate sexually.

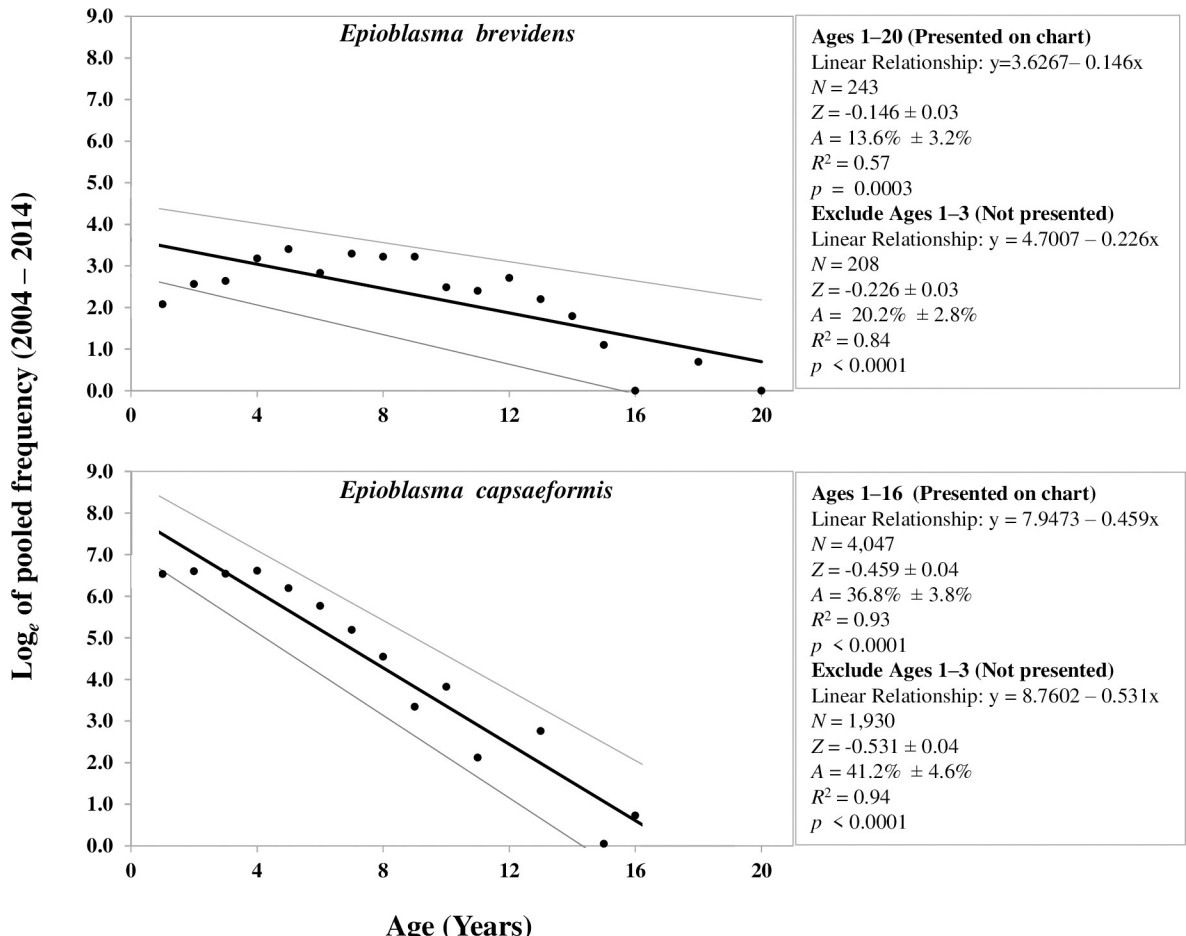

**Fig 8. Mortality rates of *Epioblasma brevidens* and *E. capsaeformis* estimated from catch-curve linear regression analyses based on age class frequencies of all individuals sampled at Swan Island, Frost Ford, and Wallen Bend in the Clinch River, Hancock County, TN, from 2004–2014.** As shown, Ages 1–3 were included in age-class frequency plots, where thin grey lines are 95% confidence intervals of the mean linear regression line (Bolded). For both species, estimated mortality rates are reported in the respective boxes to the right of the age-class frequency plots, where $N$ = sample size, $Z$ = instantaneous mortality rate, and $A$ = annual mortality; rates were estimated with and without Ages 1–3. Age 0 individuals were not used in the analyses.

From 2004–2014 for the three sites combined, mean annual population growth rate of *E. capsaeformis* was 6.3% and total population growth was 84%. Despite variation in the magnitude of population size observed between sites, population growth was similar among sites. Annual growth was 6.5% at SI, 6.4% at FF, and 4.9% at WB, while total growth was 88% at SI, 85% at FF, and 61% at WB. The variance of the mean realized per-capita growth rates at SI ($\bar{r}$ = 0.06; $\hat{\sigma}^2$ = 0.16), FF ($\bar{r}$ = 0.06; $\hat{\sigma}^2$ = 0.32), WB ($\bar{r}$ = 0.05; $\hat{\sigma}^2$ = 0.19), and the total combined (pooled) population ($\bar{r}$ = 0.06; $\hat{\sigma}^2$ = 0.16), were more than twice their mean (Table 4). Recruitment of Age-1 *E. capsaeformis* ranged from 4.3% in 2004 to 49.9% in 2007. Mean recruitment estimated across all years and sites was 20.0% (SE = 2.6). During this period, recruitment was variable, while population size of adults remained relatively stable (Table 4; Fig 6; S1 File). Following low recruitment observed in 2010, annual recruitment remained stable until the conclusion of the study, with no significant increases or decreases. There was a significant positive relationship ($p$ = 0.0005, $R^2$ = 0.75959) between the $\text{Log}_e$-transformed number of adults and $\text{Log}_e$-transformed number of recruits in *E. capsaeformis* over the study period [$ln$(Recruits) = -4.6754+1.4944$^*ln$(Adults)]. For ages 1–16, mean annual mortality for *E. capsaeformis* was

36.8% (SE = 3.8%), and when ages 1–3 were removed from the analysis, annual mortality increased to 41.2% (SE = 4.6%) (Fig 8).

Estimates of population growth for *E. brevidens* and *E. capsaeformis* based on adults only, i.e., by removing the 1–3 years olds from the analysis, resulted in quite different estimates (Table 4). The one exception was *E. brevidens* at SI, where juveniles did not influence annual or total population growth, which stayed the same at -6.7% annual growth and -50% total growth. In contrast, at FF the adult only population of *E. brevidens* experienced 24.6% annual growth for a total growth of 800% over the study period. At WB, the adult *E. brevidens* population experienced 14.1% annual growth and a total of 275% growth over the study period. The annual growth (12.0%) and total growth (212%) for the adult only population also were higher for the total pooled population of the species.

For *E. caspaseformis*, the adult only population was more stable than when juveniles were included in the estimates. At SI, annual growth was slightly less at 5.6% and the total growth was 73% for the study period. Similarly, at FF, the annual growth was 3.7% and the total growth was 44% over the entire study period. At WB, annual growth was 0.5% and total growth was only slightly positive at 5% for the study period. The total pooled population experienced 3.7% annual growth and 43% total growth, each about half of the population growth calculated from including juveniles in the estimates.

## Female fecundity

Mean total length and estimated age of female *E. brevidens* sampled for fecundity (number of glochidia per female) was 46.4 mm (range 42.1–52.2 mm) and 8.4 years (range 6–13 years). Observed mean fecundity of females ($N$ = 15) was 34,947 (SE = 2,492) glochidia, and ranged from 18,987–56,151 (Table 5; S4 File). Mean total length and estimated age of female *E. capsaeformis* was 39.7 mm (range 35.0–46.1 mm) and 5.7 years (range 4–9 years). Observed mean fecundity of females was 9,558 glochidia (SE = 1,460) and ranged from 3,456–22,182 (Table 5; S5 File).

From previously sampled females of *E. capsaeformis* obtained in spring 2002 from the Clinch River [4], mean reported fecundity was 13,008 (SE = 1,460) glochidia per female, with a minimum of 7,780 and a maximum of 16,876. Individuals from this sample ($N$ = 10) had a mean length and age of 41.5 mm (range 36.7–46.4 mm) and 6.5 years (range 5–10 years). Thus, mean total length and age of female *E. capsaeformis* pooled from 2002 [4] and 2013 was 40.3 mm (range 35.0–46.4 mm) and 6.0 years (range 4–10). No significant difference was detected between samples in 2002 and 2013 for mean length ($p$ = 0.1334), mean age ($p$ = 0.2027), or mean fecundity ($p$ = 0.1238) (S6 and S7 Files). Mean fecundity in the pooled sample ($N$ = 30) was 10,708 (SE = 785) glochidia (range 3,465–22,182) (Table 5; S6 File). Mean fecundity of *E. brevidens* and *E. capsaeformis* were statistically different ($p$<0.0001) (S6 File).

Positive curvilinear relationships were detected between both length and estimated age and female fecundity in samples of both species. Regression lines fit to the data revealed significant power-curve relationships between length and fecundity as well as second-order polynomial relationships with respect to age as a predictor of fecundity in both species (Fig 9). The relationship between transformed (Log$_e$) shell length of *E. brevidens* showed a significant positive linear relationship with transformed (Log$_e$) fecundity [$y$ = 2.8119$x$–0.3595; $R^2$ = 0.463; $p$ = 0.0057] (S7 File). Similarly, the transformed age values of *E. brevidens* revealed a significant positive linear relationship with transformed fecundity [$y$ = 0.8489$x$+8.6436; $R^2$ = 0.548; $p$ = 0.0006] (S7 File). The transformed (Log$_e$) shell length of *E. capsaeformis* also showed significant positive linear relationship with transformed (Log$_e$) fecundity [$y$ = 3.6945$x$–4.4478; $R^2$ = 0.436; $p$<0.0001] (S7 File). Likewise, the transformed age values of *E. capsaeformis* showed a

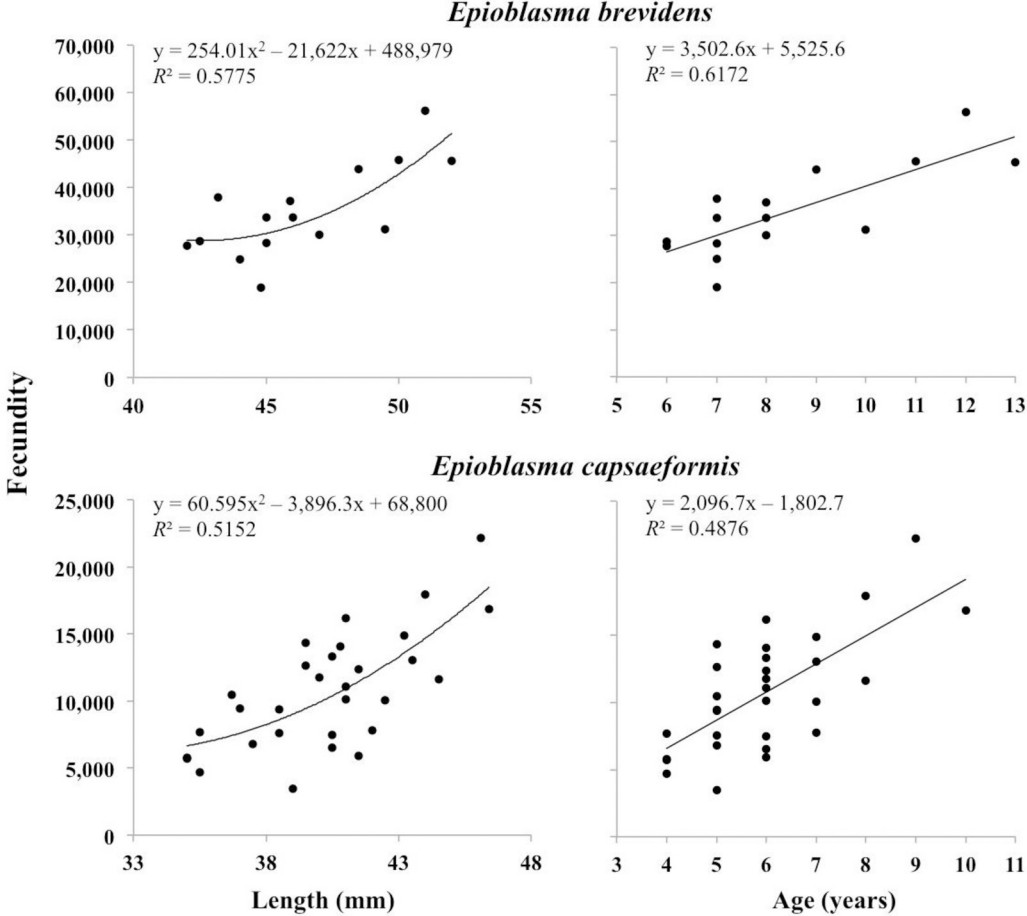

**Fig 9. Length–and age–fecundity relationships observed for _Epioblasma brevidens_ (length: _p_ = 0.0057; age: _p_ = 0.0006) and _E. capsaeformis_ (length: _p_<0.0001; age: _p_<0.0001) sampled at Kyles Ford in the Clinch River, Hancock County, TN, in 2013.** Ages were estimated based on shell length using von Bertalanffy growth curve equations presented in (4). Data for _E. capsaeformis_ includes 10 additional samples collected from various sites in the Clinch River, Hancock County, TN, in 2002 [10,23].

significant positive linear relationship with transformed fecundity [$y = 1.2239x + 7.0406$; $R^2 = 0.438$; $p < 0.0001$] (S7 File).

By extrapolation, the estimated number of glochidia brooded by female _E. brevidens_ over the study period varied dramatically among sites and was directly determined by the proportion of mature adults in the population across census years (Fig 6; Tables 3 and 4). Total glochidia brooded annually by female _E. brevidens_ ranged from 5.3E+6 to 6.2E+7 at SI, 2.5E+7 to 1.6E+8 at FF, and 2.1E+6 to 1.7E+7 at WB (Fig 10). The number of glochidia brooded by female _E. capsaeformis_ among sites ranged from 2.2E+6 to 5.2E+7 at SI, 3.1E+8 to 1.5E+9 at FF, 1.4E+7 to 7.2E+7 at WB (Fig 10). The (Log$_e$) number of glochidia brooded by females per year and the (Log$_e$) number of recruits in subsequent years showed non-significant positive correlation for both _E. brevidens_ ($R = 0.499$; $p = 0.1422$) and _E. capsaeformis_ ($R = 0.526$; $p = 0.1185$).

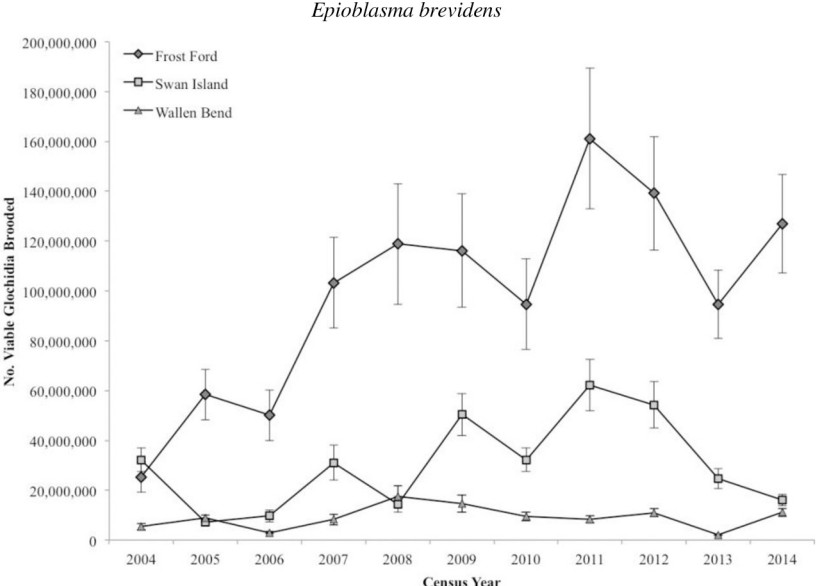

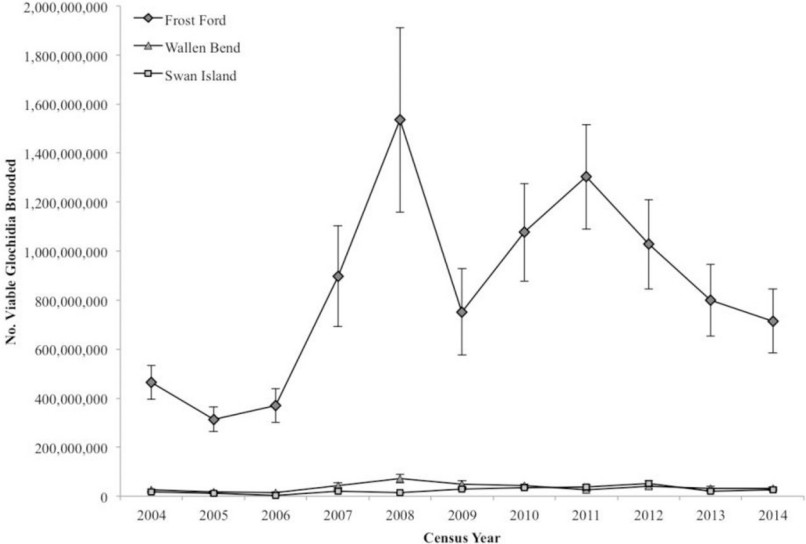

**Fig 10. Estimated number of glochidia (larvae) annually brooded by female *Epioblasma brevidens* and *E. capsaeformis* and available to host fishes at Swan Island, Frost Ford, and Wallen Bend in the Clinch River, Hancock County, TN, from 2004–2014.** Numbers were estimated based on annual number of mature females per site and the fecundity-length relationships in Fig 9.

### Influence of stream discharge on population growth

Stream discharge in the Clinch River was highly variable over the study period, with discharge lowest prior to 2007 and 2008 censuses (Fig 11). In the summer of 2007, historic low recorded-discharge was observed in the Clinch River (100–150 cfs). Further, in the years 2005–2008 and 2014, maximum-recorded discharge did not exceed 15,000 cfs, as spring flooding was less severe than in other years (Fig 11).

An IHA analysis comparing the study period to the preceding 74 years provided some evidence for differences in discharge but little coherent directionality, so differences were most

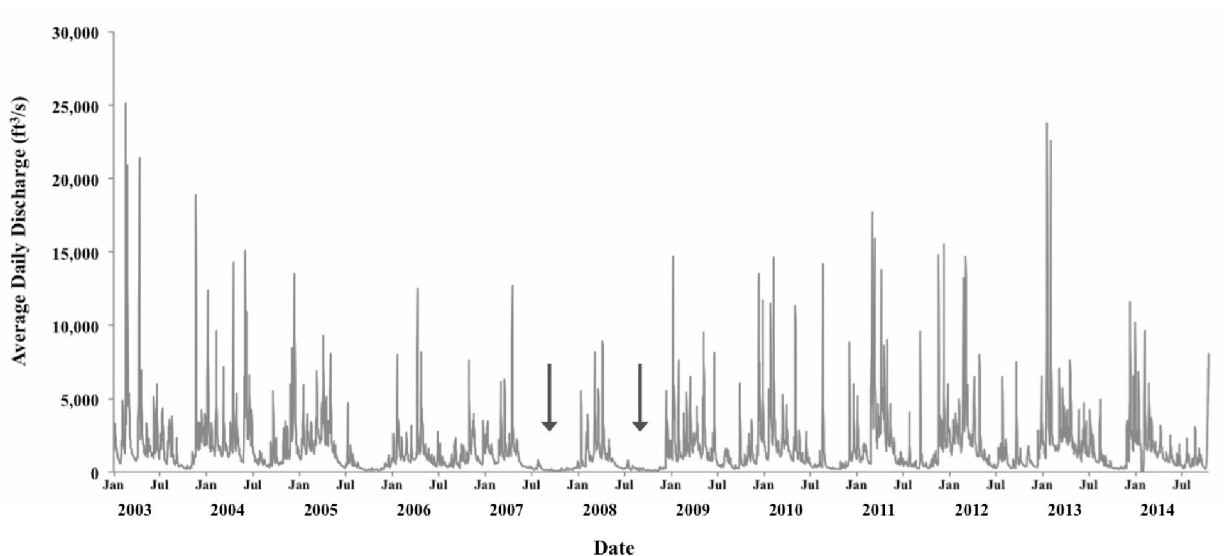

**Fig 11. Stream discharge (ft³/sec) recorded in the mainstem Clinch River above Tazewell, Claiborne County, Tennessee (data from U.S. Geological Survey, gauge 03528000) from June 2003 to December 2014.** Arrows indicate periods of low discharge conditions in the river from early spring to early winter.

likely the product of comparing a small dataset to a much larger one. Nonetheless, the study period stands out for having an extended period of extreme low discharge and a low occurrence of floods. Discharge preceding 2007 and 2008 censuses were unusual for both the study period and the 74 preceding years (Fig 11). First, 2008 had a high number of days with extremely low discharge (130 days with cfs ≤259, which was the 10th percentile of discharge). This was the greatest number of days classified as extremely low discharge in any of the 84 years. Second, both 2007 and 2008 had low discharge from 1 June to 30 September, with each of those months below or at 10th percentile for monthly statistics derived using all 84 years of data in IHA. Third, the year proceeding 2008 census had lowest annual median discharge, 360 cfs, which was 41% of the next highest value during the study period and lowest of the 84 years in the dataset. The study period was also notable for the rarity of flood events. The IHA analysis demonstrated that only one flood event occurred in the study period (prior to the 2013 census), whereas 40 of the previous 74 years had at least one flood event classified as either small, large, or both. The highest mean daily discharge of the study period was 32,100 cfs. The highest mean daily discharge in the preceding 74 years was 83,300 cfs—2.6 times greater. These observations were used to guide selection of discharge statistics and hypotheses.

Just over half of 190 Pearson Product-Moment Correlations (Pearson's r) pairwise comparisons of discharge statistics were moderately to strongly correlated at α <0.1 (S8 File). Almost a fourth of the pairwise comparisons were strongly associated, having a Pearson's r >0.7. If random, 10% of pairs would be expected to be correlated at α <0.1, with few apparent patterns. Correlations among many discharge statistics suggested many would have comparable associations with any dependent population parameter, thus necessitating careful consideration of inclusion in hypotheses and model comparisons. Correlations between discharge statistics and realized population growth indicated relationships were highly dependent on the inclusion of individuals classified as juveniles (S9 File). When only adults were used to calculate population growth, no discharge statistics were correlated with realized *E. brevidens* population growth (at α <0.1), and only 1 of 20 discharge statistics was associated with *E. capsaeformis*. When juveniles were used to calculate realized population growth 9 of 20 pairs

**Table 6. Linear models explaining realized per-capita growth rate for *Epioblasma brevidens* and *E. capsaeformis* using flow statistics were compared by Akaike Information Criteria (AICc).** Growth rate was calculated using juveniles and adults. ΔAICc >2 is generally accepted to be a model with greater relative support. Small sample sizes limited the range of responses. k = number of parameters in a model.

AIC model scores for *Epioblasma brevidens*:

| Flow Statistic | k | AICc | ΔAICc |
|---|---|---|---|
| Median Mean Daily Discharge for August | 2 | 10.842 | 0.000 |
| Median Mean Daily Discharge for Year | 2 | 12.498 | 1.656 |
| Median Mean Daily Discharge June-September | 2 | 12.612 | 1.769 |
| 90 Day Minimum Flow | 2 | 12.659 | 1.816 |
| Days Considered Low Flow in a Year | 2 | 13.905 | 3.063 |
| Days Below Median Flow (1931–2014) in a Year | 2 | 14.259 | 3.416 |
| **Null** | **1** | **14.666** | **3.824** |
| 90 Day Maximum Flow and 90 Day Minimum Flow | 3 | 16.451 | 5.608 |
| 90 Day Maximum Flow | 2 | 16.815 | 5.972 |
| Days Considered Extreme Low Flow in a Year | 2 | 16.844 | 6.001 |
| Days Considered Extreme High Flow in a Year | 2 | 16.998 | 6.156 |
| Days Considered Extreme Low Flow and Extreme High Flow in a Year | 3 | 22.522 | 11.680 |

AIC model scores for *Epioblasma capsaeformis*:

| Flow Statistic | k | AICc | ΔAICc |
|---|---|---|---|
| Median Mean Daily Discharge for August | 2 | 13.372 | 0.000 |
| Days Considered Extreme High Flow in a Year | 2 | 17.610 | 4.238 |
| Days Considered Low Flow in a Year | 2 | 18.248 | 4.876 |
| 90 Day Minimum Flow | 2 | 18.331 | 4.959 |
| Median Mean Daily Discharge June-September | 2 | 19.059 | 5.687 |
| Median Mean Daily Discharge for Year | 2 | 19.170 | 5.798 |
| Days Below Median Flow (1931–2014) in a Year | 2 | 19.326 | 5.954 |
| **Null** | **1** | **19.546** | **6.174** |
| Days Considered Low Flow in a Year | 2 | 20.859 | 7.487 |
| 90 Day Maximum Flow | 2 | 20.881 | 7.509 |
| 90 Day Maximum Flow and 90 Day Minimum Flow | 3 | 24.041 | 10.668 |
| Days Considered Extreme Low Flow and Extreme High Flow in Year | 3 | 24.066 | 10.694 |

and 10 of 20 pairs were correlated, respectively. This analysis led to the inclusion of a select number of discharge statistics in models (Table 6).

*Epioblasma brevidens* realized per-capita growth rate was negatively associated with median discharge especially during summer (June-September). Other models that focused on measures of low discharge tended to perform better than the null model and far better than models that included statistics describing high discharge. Because it was the most intuitive, the model where *E. brevidens* realized population growth was negatively associated with annual median discharge was presented (Fig 12; y = 0.7665–5.8E$^{-5}$*Median Year, F = 7.2532; d.f. = 1,8; *p* = 0.0274) along with the best supported model, which suggested flows in August were important (y = 0.7174–1.1E$^{-3}$*Median August, F = 9.999; d.f. = 1,8; *p* = 0.0133). Coincidentally, the lowest population growth values were observed when the singular flood event occurred in the study period (2013).

A greater number of models better approximated the relationship between discharge and realized per-capita growth rate for *E. capsaeformis*. Support ranged from marginal to relatively strong given limited sample size. As with *E. brevidens*, there was evidence that August median discharge was important (Fig 13, y = 0.9566–1.5 x 10$^{-3}$*Median August, F = 14.7688; d.f. = 1,8;

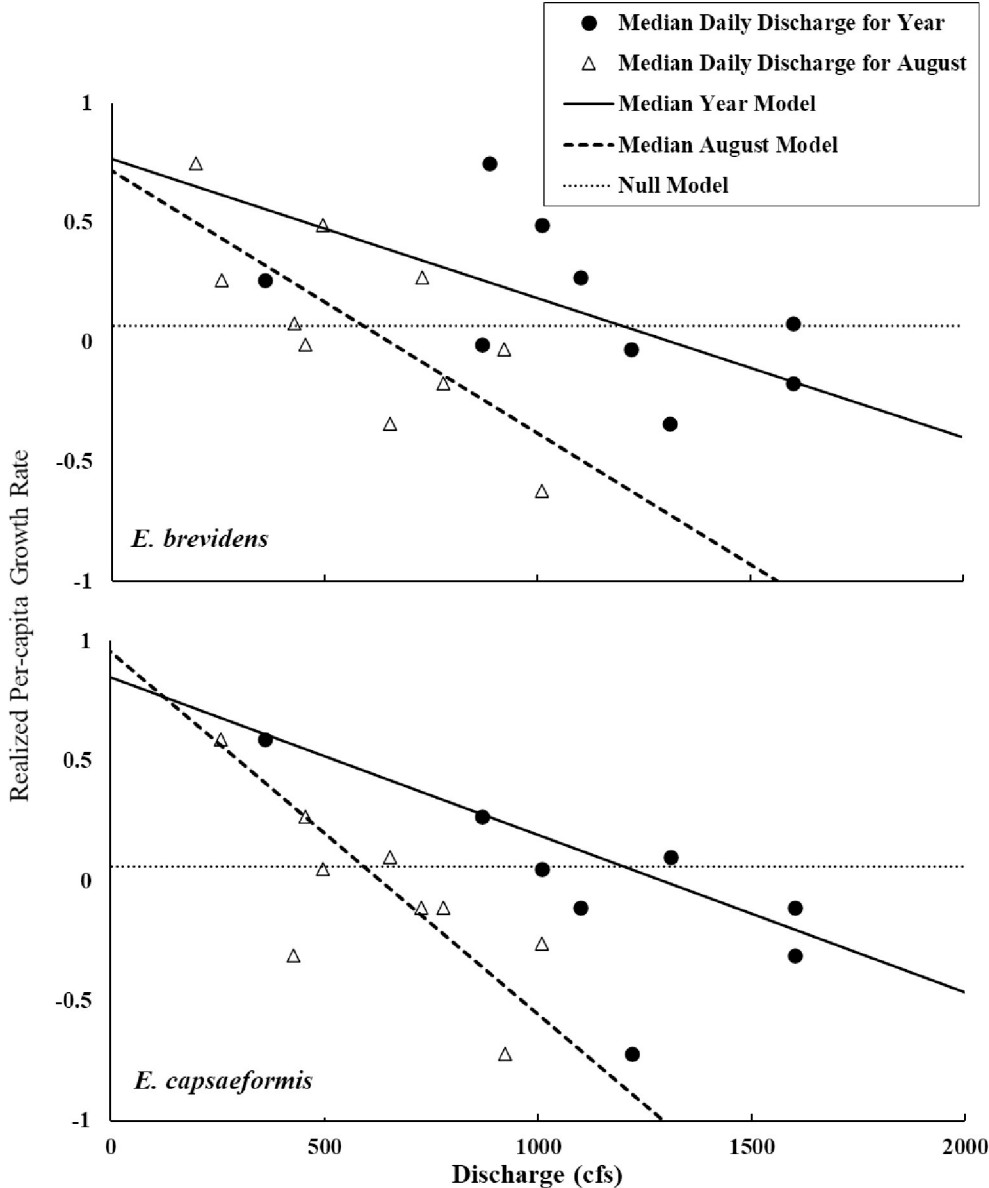

**Fig 12. Median daily discharge for each year and for August were negatively associated with realized per-capita growth rate for *Epioblasma brevidens* (top) and *E. capsaeformis* (bottom). Growth rate was calculated using both juveniles and adults combined.**

$p = 0.0049$). For comparison, the relationship between realized per-capita growth rate for *E. brevidens* and median daily discharge for each year was plotted (y = 0.8502–6.6 x $10^{-5}$*Median Year, F = 4.7507; d.f = 1,8; $p = 0.0609$). The model demonstrating a negative association between extreme high discharge and realized per-capita growth rate for *E. capsaeformis* also had strong support (Fig 13, y = 0.7997–0.02238*Days Extreme High Discharge, F = 6.1542, d.f. = 1,8; $p = 0.0381$).

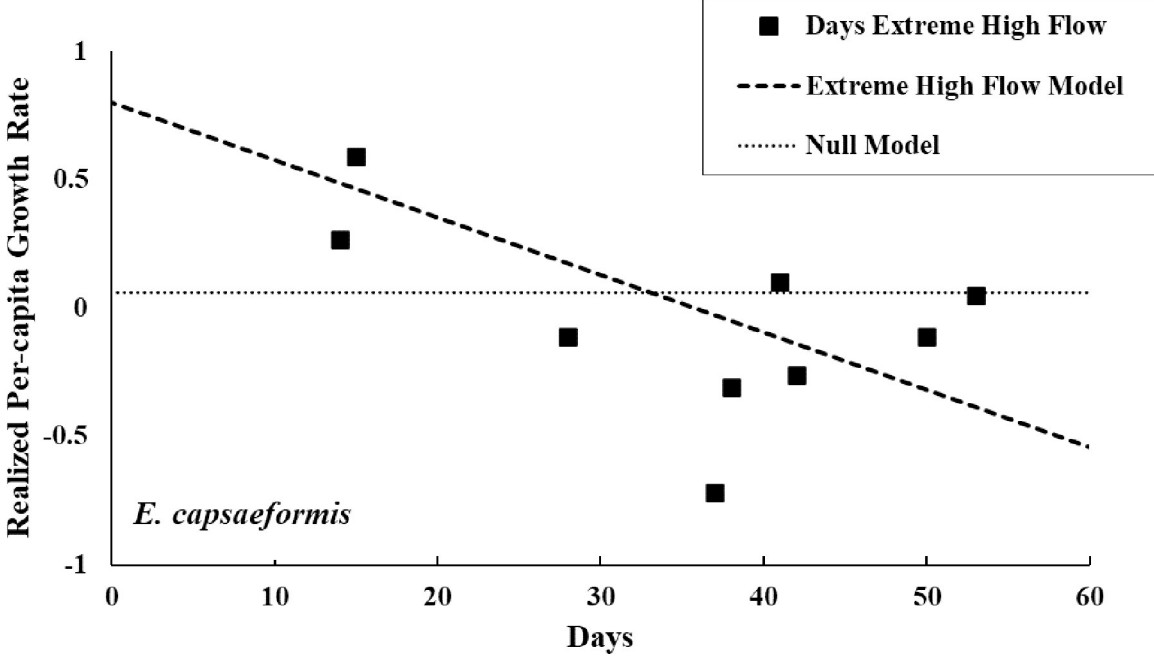

**Fig 13. The number of days with extreme high discharge ($\geq$ 90th percentile of flows 1931–2014) in a year was negatively associated with realized per-capita growth rate for *E. capsaeformis*.** Growth rate was calculated using juveniles and adults.

## Discussion

### Spatiotemporal variation and the influence of stream discharge on demographic responses

This study found evidence that stream discharge was associated with fluctuations in population size of *E. brevidens* and *E. capsaeformis* and other demographic parameters such as recruitment of juveniles and population growth. Further, statistical associations between these parameters and discharge were stronger when juveniles were included analyses, and weaker for the adults only. Lower discharge in the late spring and summer likely benefits adult reproduction and juvenile survival and growth in several key ways. Gentler flow conditions allow for more interaction between hosts fishes and female mussels, thus increasing the rate of glochidial parasitism [4]. Typically, lower flows are less turbid, so more sunlight to the stream bottom of shallow shoal areas and in deeper pools can increase primary productivity, which may in turn increase food availability to juvenile mussels [24]. And perhaps the biggest benefit, stable habitat conditions allow juveniles to survive and grow instead of constantly being disturbed or even displaced into unfavorable habitats from benthic scouring during high discharge events [4].

However, there are two major caveats for interpreting associations between discharge and realized population growth. First, are the associations an artifact of sampling or are they biologically significant? And second, were appropriate measures of variation selected? Datasets were re-examined to separate-out the difference between how discharge influenced detection of juveniles (sampling) versus how discharge influenced demographic parameters (biological significance). A *post hoc* comparison using Pearson's r suggested discharge in the days prior to sampling—calculated as median discharge for 30 D, 60 D, and 90 D prior sampling—was associated with the proportion of juvenile *E. brevidens* in a census but correlations with realized population growth were weak (S8 and S9 Files). Correlations were even weaker between the

portion of *E. capsaeformis* juveniles in a census and discharge in days prior to sampling. These results suggested that discharge statistics were more likely associated with demographic parameters rather than the timing of censuses. As for the second caveat, the IHA analysis produced hundreds of discharge statistics and many were correlated. The final set of correlations examined were selected based on experience and relative ease of interpretation. Moreover, correlation is likely to increase with a greater study period, so no single statistic, association or model examined should be considered definitive. These are hypotheses to be reexamined with more data and by other researchers.

The population of *Epioblasma brevidens* remained relatively stable over the 11 census years (Fig 2), exhibiting some variability attributable to discharge. However, the high level of sampling variation due to the species' low occurrence density ($<1$ m$^{-2}$) observed across time intervals made detecting and analyzing patterns difficult. Hence, using trends in population size to determine the species' long-term population viability may prove challenging even in the future as more data is collected. The proportion of juveniles to adults showed how this species may benefit from periodic low discharge conditions, which presumably increases recruitment of juveniles. It is important to note that the population size of adults was far less variable than the total population size (including sub-adults and juveniles) over the study period (Fig 6; Table 3). Similar to findings of [25], young individuals comprised multiple age classes in most censuses, indicating recruitment occurred on a regular basis. Although an increasing trend in population size from 2008–2011 was observed in *E. brevidens*, by the end of the study the population had returned to population sizes similar to those observed at the beginning of the study period or were trending in that direction. For instance, at WB, population size in 2004 was estimated to be 636 individuals, and by 2008, the population had grown to an estimated 2,121 individuals but by 2014 was back to an estimated 796 individuals. A closer look at juvenile population sizes in each census compared to adults only, shows that the increased population size during the first half of the study period was almost entirely represented by smaller age classes. So, it is likely only a small proportion of those survived to adulthood (Table 3; Fig 6). This result is important because a less exhaustive monitoring study, e.g., over a shorter study period, might have given a false sense of the directionality of either a positive or negative trend for this population. This study illustrates how important it is for mussel populations, in particular adult populations, to be studied over long time periods (10 or more years) so all of the fluctuations can be put into context. Further, quantitative surveys need to explicitly assess quadrat sample size for lower density ($<1$ m$^{-2}$) species like *E. brevidens*. If a quadrat sampling design cannot factor-out real temporal change from overwhelming sampling variation, than another sampling methodology such as Capture Mark-Recapture (CMR) sampling might be a better choice. In general, the adult population was quite stable from 2004–2014 and neither greatly increased or decreased relative to any one census. This has been reported in other mussel populations, in the absence of human perturbations [26–28]. The low level of observed temporal (process) variation indicated that the population was less variable than initially observed from plotting the raw data. Temporal variation at FF ($C_v(T) = 0.221$) was nearly half that of total variation in the trend ($C_v(S) = 0.397$), where the largest population size was observed. However, the temporal variation at SI ($C_v(T) = 0.413$) and WB ($C_v(T) = 0.444$) were nearly twice that of FF, indicating that the FF deme was less variable (Fig 2; S1 File). The larger population size at FF may have contributed to this stability, making it more robust to environmental and demographic fluctuations.

When density was $>0.6$ m$^{-2}$ at each site and when pooled, the finite rate of population growth of *E. brevidens* in the subsequent time step always dropped below 1, indicating juvenile recruitment was regulated by biotic and abiotic factors, essentially the population would hit a ceiling (Table 3; Fig 2). A review of all available density data for the species from 1979 to 2014

collected in the Clinch River, found only 9% of observations exceeded 0.6 m$^{-2}$, with the highest being 1.1 m$^{-2}$ (S10 and S11 Files). Further, each time densities were <0.3 m$^{-2}$, population growth always increased, exhibiting annual increases as high as 400%, for example at WB from 2006–2007 (Table 3). From 2007–2009 when population growth was highest at FF, population growth at WB and SI also was highest but at even higher rates (Fig 2). Further, as population size declined at FF and WB from 2008–2009, population size downstream at SI increased, perhaps due to fish migration between shoals or possibly downstream migration of mussels during high flow events.

In contrast to *Epioblasma brevidens*, the population of *E. capsaeformis* was highly variable over the 11 censuses, but was more stable in the second half of the study period (Fig 3). The population exhibited a strong response to variation in stream discharge, resulting in higher juvenile recruitment and population growth rate in years of low spring-summer discharge (Tables 2 and 4). This phenomenon is not unprecedented in other free-flowing streams outside of the Clinch River [29,30], which have also showed high and variable recruitment in particular species during low-flow years. The low level of sampling variation observed across all time intervals suggests that these demographic responses to environmental fluctuations were real and linked to the species' long-term population viability. Further, the amount of temporal (process) variation in population size suggests that the population is naturally highly variable and at a higher risk of decline and extirpation due to environmental stochasticity in the CR [31,32]. Temporal variation at FF ($C_v(T)$ = 0.541) was not that different from total variation observed in the overall trend in population size ($C_v(S)$ = 0.551), where the largest population size was observed. However, the temporal variation at SI ($C_v(T)$ = 0.684) and WB ($C_v(T)$ = 0.589) were higher than that of FF (Fig 3; S3 File). In comparison to *E. brevidens* at the same sites, *E. capsaeformis* was much more variable and influenced by environmental and demographic variation. These findings may help to explain how populations of *E. brevidens* have remained more viable in the river and across its range than have those of *E. capsaeformis*.

Variation in stream discharge affected *E. capsaeformis* population growth in two apparent ways. First, extended periods of low discharge (days not exceeding the 10th percentile observed for the study period) occurred simultaneously with a large increase in positive population growth (Figs 3 and 11). Further, in years that exhibited more recorded discharges that exceeded the 90th percentile observed for the study period, adult mortality increased or was not supplemented with equal levels of juvenile recruitment (Figs 3 and 12). Given its small body size, these findings illustrate the vulnerability of *E. capsaeformis* to stream discharge periods of severe flooding. Further, at FF the population exhibited positive growth at densities ~20 m$^{-2}$ on three occasions (from 2007–2008, 2009–2010, and 2010–2011). Above that density, the population at FF exhibited negative growth, which may indicate a ceiling or carrying capacity at this site at <40 m$^{-2}$. At SI and WB, the population ceilings appeared to be lower (5–10 m$^{-2}$) compared to FF (Fig 3).

## Age structure and demographic vital rates

Age structure of *E. brevidens* was similar across all three study sites from 2004–2014 and was comprised mostly of adults (Figs 4 and 6). The oldest individual sampled was estimated to be 20 years old based on its shell length but, a 28-year-old individual was recorded in the Tennessee section of the river [4]. So, it is likely that pulses in recruitment observed in the population could have lasting effects of up to 20 years or longer. In 2007 and 2008, the population had a higher number of recruits compared to other census years, which suggests that the population was influenced by low-flow conditions in previous years prior to the initiation of the study. Length frequencies and longevities of *E. brevidens* were positively biased towards males (Fig

5), which agrees with the biology of the species, where females halt shell growth after reaching maturity and put more energy into gamete production and fish host attraction. Releasing glochidia contributes to females' susceptibility to substrate abrasion and dislodgement during high flows and vulnerability to predation, and thus ultimately to higher mortality, as they sometimes do not completely burrow. Lower observed maximum age of females and their skewed sex ratio are likely linked to these reproductive behaviors (Fig 7).

Age structure of *E. capsaeformis* was similar across all study sites from 2004–2014, which was comprised of juveniles and sub-adults with fewer older individuals (Figs 4 and 6). At all sites the frequency of 1-year-olds was proportionally higher from 2006–2008 and noticeably less in 2009, which corresponds to the high population growth observed during those years. This high population growth strongly coincided with periods of low discharge during spring through autumn prior to each of these censuses. The oldest individual sampled across all census years was 16-years-old based on shell length but a 12-year-old individual was aged from shells [4]. It is possible that individuals of *E. capsaeformis* in the Clinch River live beyond this age, since individuals of the species as old as 17 years have been observed in the nearby Nolichucky River, a stream also located in eastern TN (T. Lane, unpublished). This is still approximately half the life expectancy of *E. brevidens*. Similar to *E. brevidens*, length frequency of *E. capsaeformis* was positively biased towards males and agrees with the biology of the species. Higher mortality of mature females is likely related to behavioral changes related to release of glochidia and host fish attraction, where they are vulnerable to predation and substrate abrasion and dislodgement during high-flow events. The sex ratio of the population also was skewed toward males, as females represented 42% of the total sample across sites and years. However, when data from 2009 was removed from the analysis, females were 45% of the total. Thus, sex ratios may remain more consistent over time for *E. capsaeformis*, which is dissimilar to the sex ratio observation of *E. brevidens*.

Annual mortality rates of both species were substantially different based on catch-curve linear regression analysis, with mortality of *E. brevidens* at 13.6% (±3.2%) per year and *E. capsaeformis* at 36.8% (±3.8%) per year. Further, when the age-frequencies were truncated (removing ages 0–3), annual mortality of *E. brevidens* increased to 20.2% (±2.8%) and that of *E. capsaeformis* increased to 41.2% (±4.6%). These findings suggest that each species has different life history strategies, where *E. brevidens* has slower body growth, lower recruitment rates, higher age at maturity, and higher adult survival compared to *E. capsaeformis*, which has faster body growth, higher recruitment rates, lower age at maturity, and lower adult survival. The data suggests then that *E. capsaeformis* needs higher and more consistent annual recruitment compared to *E. brevidens* to maintain a viable population [4]. This higher recruitment need is further reflected in the preferred fish hosts of each species, where *E. capsaeformis* uses shorter-lived and locally abundant *Etheostoma* darter species, while *E. brevidens* uses longer-lived but potentially more mobile and less abundant *Percina* darter species.

Fecundity of females was much higher in *E. brevidens* compared to *E. capsaeformis*. In fact, the average number of glochidia in *E. brevidens* (34,947 CI±4,884) was more than three times the average of *E. capsaeformis* (10,708; CI±1,539) (Table 5). The length- and age- relationships to fecundity were both positive, a relationship that has been observed in numerous mussel species [33–35]. Interestingly, the relationship for length followed a polynomial trend, while age was linear (Fig 9). The higher fecundity of *E. brevidens* presumably allows for more glochidia to be released to and encysted on a larger bodied but less abundant fish host, to increase recruitment of juveniles. Thus, recruitment is being sustained utilizing less abundant hosts but those likely parasitized with a greater number of glochidia. Further, it must trade-off slower body growth to higher energy allocation to fecundity. In contrast, *E. capsaeformis* utilizes much more abundant fish hosts that are presumably encountered and infected more

frequently. By putting less energy into gamete production, it can expend more energy on body growth, perhaps further aiding its ability to successfully parasitize its host(s) as it ages.

When fecundity data were extrapolated to the population size estimates of adult females, it became clear that *E. capsaeformis* had much more available glochidia from year to year than *E. brevidens*, most noticeably at FF. There, at its peak in 2011, *E. brevidens* was estimated to have 1.5E+8 glochidia brooded by females and potentially available to host fish (Fig 10). This seems high when considering only thousands of juvenile recruits were estimated in the study locations from year to year. So, it may indicate that the luring activity of the species is less efficient or, as stated earlier, that infected fish hosts are migrating away from the site and thus any potential recruits are excysting and settling outside of the main study sites. In contrast, *E. capsaeformis* at its peak in 2008 was estimated to have roughly 1.5E+9 or nearly 10 times this number of glochidia available to host fish (Fig 10) compared to *E. brevidens*. Contrastingly, tens of thousands and even hundreds of thousands of these glochidia were able to fully develop into juveniles and thus be observed locally as recruits in the subsequent censuses (Table 3; Fig 6). This finding agrees with our conclusions that *E. capsaeformis* demonstrates more R-selected fecundity traits relative to *E. brevidens*. With many fewer adult years to be expected, *E. capsaeformis* demonstrates a need to maintain larger population size of brooders in any given year who must contribute recruits early and often to ensure stable population growth.

## Influence of fish hosts on mussel population demography and life history

This study suggests *Epioblasma brevidens* is best characterized as a periodic strategist as hypothesized by [28], which has been characterized by moderate to high body growth, intermediate life span, earlier maturation, intermediate fecundity, and smaller body size relative to other North American mussel species. This life history strategy is intermediate between the *r*- and *K*- selection continuum [34,35]. However, the extreme mortality of glochidia, lower annual recruitment of juveniles, and higher annual survival of adults questions the strict placement of *E. brevidens* into this continuum. Its life-history strategy may simply reflect in part the population dynamics and biology of its preferred host fishes. These life-history strategies and the demographic data of this study help to explain how this species has persisted in multiple fragmented populations in the Clinch River and other parts of its known range.

In contrast, this study suggests *Epioblasma capsaeformis* is best characterized as an opportunistic strategist as hypothesized by [28], which has been characterized by high body growth, short life span, early maturation, and smaller body size relative to those of other freshwater mussels. This life-history strategy is closer to the *r*-selection continuum, than are the periodic or equilibrium strategies. Still, *E. capsaeformis* is longer-lived and has lower fecundity than most species that can be grouped into this type of strategy, which may allow it to persist for extended periods of time when conditions are unsuitable for recruitment. High annual recruitment of early life stages to adult stages and lower annual adult survival suggests that it strongly depends on the presence of a consistently high number of local fish hosts and is overall more susceptible to natural and/or anthropogenic-influenced fluctuations to its environment. This helps explain the species' inability to persist across much of its historical range, where fragmentation, flow regime alteration, pollution events, and habitat degradation likely influenced more rapid population extirpation than would have been experienced by proposed periodic or equilibrium strategists.

Specialization of morphologically and behaviorally complex mantle lure strategies (like other *Epioblasma* spp.) suggests each species utilizes specific host fishes. Fishes in the genus *Etheostoma* captured by *E. brevidens* in laboratory trials were killed as the result of their skulls being crushed, while *P. caprodes* (having a sturdier skull and broader frontal bones) was more

likely to survive the encounter [11]. Therefore, even if small *Etheostoma* spp. can successfully transform juveniles of *E. brevidens*, they may not be the primary host under natural conditions. So, presence of particular fish species at a recipient site for stocked mussels may not necessarily guarantee its suitability as a host regarding minimum recruitment rates. The host-parasite dynamics of freshwater mussels and their respective fish hosts undoubtedly is a main driver of mussel population dynamics [36,37], as it has been shown that parasites act as strong regulators of host populations and parasite-free host populations are likely to exhibit altered population dynamics when compared with host populations in the core of a parasite's range [38,39]. For example, this could influence recruitment rates of reintroduced mussel populations as they are stocked in locations where host populations have not been subjected to the parasitic larval stages of a mussel species for multiple generations. This could increase or even hinder glochidial transformation and recruitment rates in reintroduced mussel populations. Further studies related to the ecology and biology of these two endangered species and other species filling similar community niches will help prepare managers as they address the recovery goals of each species.

## Summary and management recommendations

This study was conducted during a unique period of stream discharge that affected populations of both mussel species in two ways: 1) the annual censuses overlapped an extreme drought that resulted in low-discharge conditions and high population growth for both species in the CR [4], and 2) populations of each species grew large enough during this period that their growth and population size was potentially governed by density-dependent factors. Each species' demographic response to normal and low stream discharge conditions was quantified during this study and now can be used to predict future population responses to discharge in and outside the study area. Whether direct or indirect, recruitment responded positively to low flow conditions in these populations during the study. This could be explained in a number of ways: 1) lower discharge could cause higher amounts of primary food to bloom and become available to early life stages of these mussels, 2) female mussel to fish host encounters may have a higher incidence when peripheral stream habitat is lessened and fish become more concentrated on mussel beds; or simply, 3) more stable micro-habitat conditions for juveniles allowed them to direct energy to growth instead of constant re-positioning [4]. Other explanations also are possible, for example, aspects of the life history and demographic fluctuations of the respective host fishes also may be a primary driver of the observed mussel population responses, but this hypothesis has yet to be tested. However, if habitat conditions and host fish populations in areas where each species is being reintroduced are similar to those of the Clinch River, reintroduced populations should exhibit similar dynamics and demographic responses. Available data are not sufficiently robust to assess the degree to which density-dependence constrains populations, but clues exist. Future studies for each of these species should consider these limitations but keep in mind that ethically, more intensive sampling may not be worth it to answer this question. Nevertheless, biologists involved in stocking individuals to sites outside the Clinch River should consider our estimated density ranges as demographic targets, establishing and maintaining site densities 0.5–1 m$^{-2}$ for *E. brevidens* and 5–10 m$^{-2}$ for *E. capsaeformis*.

## Supporting information

**S1 File. Age class frequencies of *Epioblasma brevidens* and *Epioblasma capsaeformis*.** (PDF)

**S2 File. Mean lengths of *Epioblasma brevidens* and *E. capsaeformis* in quadrat samples.**
(PDF)

**S3 File. Estimated variances and population size.**
(PDF)

**S4 File. Total number of glochidia brooded per gravid female *Epioblasma brevidens*.**
(PDF)

**S5 File. Total number of glochidia brooded per gravid female *Epioblasma capsaeformis*.**
(PDF)

**S6 File. Fecundity as number of glochidia in female *Epioblasma brevidens* and *E. capsaeformis*.**
(PDF)

**S7 File. Linear relationships between natural logarithm (*Ln*) shell length and *Ln* age and fecundity in *Epioblasma brevidens* and *E. capsaeformis*.**
(PDF)

**S8 File. Pearson Product-Moment Correlations.**
(PDF)

**S9 File. Pearson Product-Moment Correlations between flow statistics.**
(PDF)

**S10 File. Plots of mussel densities of each species between 1973 and 2014.**
(PDF)

**S11 File. Data sources for mean density plots shown in S10 File.**
(PDF)

## Acknowledgments

We thank Steve Ahlstedt, Braven Beaty, Don Hubbs, Craig Walker, and the many other individuals who assisted with the long-term site monitoring in the Clinch River, TN conducted by the U.S. Fish and Wildlife Service (USFWS), the Tennessee Wildlife Resources Agency, the Virginia Department of Wildlife Resources (VDWR), The Nature Conservancy, and Virginia Tech. The views expressed in this article are those of the authors and do not necessarily represent those of the USFWS or VDWR.

## Author Contributions

**Conceptualization:** Jess Jones, Brett Ostby, Robert Butler.

**Data curation:** Tim Lane, Brett Ostby.

**Formal analysis:** Tim Lane, Jess Jones, Brett Ostby.

**Funding acquisition:** Jess Jones, Robert Butler.

**Investigation:** Tim Lane, Jess Jones, Brett Ostby.

**Methodology:** Tim Lane, Jess Jones, Brett Ostby.

**Project administration:** Jess Jones, Robert Butler.

**Supervision:** Jess Jones.

**Validation:** Jess Jones, Brett Ostby.

**Visualization:** Tim Lane, Jess Jones, Brett Ostby.

**Writing – original draft:** Tim Lane.

**Writing – review & editing:** Tim Lane, Jess Jones, Brett Ostby, Robert Butler.

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
