## [Decision Letter · Decision Letter 0]

6 Jan 2021

PONE-D-20-36478

Long-term Monitoring Data for Two Endangered Freshwater Mussel Species (Bivalvia: Unionidae) Reveal How Demographic Vital Rates Are Influenced By Species Life History Traits

PLOS ONE

Dear Dr. Jones,

Thank you for submitting your manuscript to PLOS ONE. After careful consideration, we feel that it has merit but does not fully meet PLOS ONE’s publication criteria as it currently stands. Therefore, we invite you to submit a revised version of the manuscript that addresses the points raised during the review process.

Because one of the reviewers was not able to return comments, I also provide comments focusing on statistical analyses. Overall, reviewer 1 felt the work will benefit researchers working on mussel studies by providing different approaches to commonly available data. Major concerns are centered around insufficient descriptions of methods (both sampling and analysis). My concern is that statistical analyses are outdated. For example, the conclusion about density dependence is most likely spurious (see my separate comments). The idea to separate the sampling error and process error is good; however, the method to separate them does not make sense (this may be coming from a lack of description). My opinion is vacillating between “Reject” or “Major Revision.”  Because of positive comments from reviewer 1 and my opinion that the analyses can be redone “easily” (based on my experience), I decided to require “Major Revision,” which will probably include re-doing statistical analyses unless convincing justifications are provided.

We look forward to receiving your revised manuscript.

Kind regards,

Masami Fujiwara, PhD

Academic Editor

PLOS ONE

2. We noted in your submission details that a portion of your manuscript may have been presented or published elsewhere. "The underlying density data from the three field sites was published in Jones et al. (2018), a pdf copy of that paper has been uploaded, but data were not used for estimation of demographic vital rates, which is the focus of the current submission." Please clarify whether this publication was peer-reviewed and formally published. If this work was previously peer-reviewed and published, in the cover letter please provide the reason that this work does not constitute dual publication and should be included in the current manuscript.

4. We note that Figure 1 in your submission contains map images which may be copyrighted. All PLOS content is published under the Creative Commons Attribution License (CC BY 4.0), which means that the manuscript, images, and Supporting Information files will be freely available online, and any third party is permitted to access, download, copy, distribute, and use these materials in any way, even commercially, with proper attribution. For these reasons, we cannot publish previously copyrighted maps or satellite images created using proprietary data, such as Google software (Google Maps, Street View, and Earth). For more information, see our copyright guidelines: http://journals.plos.org/plosone/s/licenses-and-copyright.

(1) You may seek permission from the original copyright holder of Figure(s) [#] to publish the content specifically under the CC BY 4.0 license. 

Additional Editor Comments:

Density Dependence

The method for detecting density dependence in the manuscript is not acceptable (lines 433-). Please read

Freckleton, R. P., A. R. Watkinson, R. E. Green, and W. J. Sutherland. 2006. Census error and the detection of density dependence. Journal of Animal Ecology 75:837-851.

The issue is that the linear regression includes the same variable (with potentially large sampling error) as a response and explanatory variables. You wrote “This relationship was strongest when population density at a site was low, as growth rate tended to increase much more rapidly in the subsequent time interval at each site” (lines 436-438). This is a strong indication that the association is spurious. The only way you can detect density dependence from the type of data you have is to use a state-space method. Please read

Lebreton, J.-D., and O. Gimenez. 2013. Detecting and estimating density dependence in wildlife populations. The Journal of Wildlife Management 77:12-23.

Separation of Process and Sampling Error (Line 173- )

“Sampling error variance was determined from the variance of the 11 annual abundance estimates. …”

This part does not make any sense to me. You only had 11 samples per cite. If you estimate the variance from 11 samples, it should be the total variance. I am speculating it is something to do with including or excluding spatial variations (among cite variations). It may also be something to do with not understanding the symbol that looks like curly “F” (line 172); it is not defined anywhere.

A standard approach to separate sampling error and process error is to use a state-space method. There are many papers on this topic, but I suggest you read the following paper:

Holmes, E. E., E. J. Ward, and K. Wills. 2012. MARSS: Multivariate autoregressive state-space models for analyzing time-series data. R Journal 4:11-19.

The paper describes a multivariate case, but it is easy to collapse down to a single variable model. The time series do not look stationary. Non-stationarity might inflate process error unless you include other processes (like density dependence or other covariates).

Estimation of Instantaneous Mortality (Line 195-)

Do you use dynamic data or static data? In other words, do you use N_0 when N_t was recruited (dynamics) or N_0 of the same year N_t was estimated (static)? In reality, the cohort that was recruited 2004 (defined as age 0) will appear as age 1 in 2005, age 2 in 2006, etc. It is not clear what is meant by “the number in a year class at time t” (line 199). Which year class? If you took the mean of all densities of the same age, I suggest you separate them. Overall, the analysis is not sufficiently described. Note there are many ways to conduct the analysis, and each has associated assumptions. It is important to clearly describe the method and justify it (this is true with all of the methods you use).

Minor Comments

Line 198: Remove parenthesis around “t”. It looks like Z is a function of t. It should be Z times t.

Line 170: clearly define the curly “F”

Line 188: define lambda_G in words later.

Line 189: lambda is the finite annual population growth rate, and r is the instantaneous annual population growth rate. Both are called population growth rates and have the unit of per year. The difference is one is a finite rate, and the other is an instantaneous rate.

Line 191: why r has a bar, and mu has hat? Remove both bar and hat. You can delete “or mu”. It does not matter what symbols others use as long as you define your parameter clearly.

Line 192: insert r bar after “the arithmetic mean”.

Line 200: “analogous”. It is a simple linear regression (not analogous). You probably do not need much explanation about the catch-curve analysis. It is a standard method in fisheries. For ecologists, you can just mention it is analogous to life-table analysis.

Line 201: “A” is the annual finite mortality rate, and “Z” is the annual instantaneous mortality rate. Both are per-capita rates so that they have the unit of per year.

Line 201: Remove round brackets around “x” in the equation. It is interpreted as b is a function of x as it is written now.

Line 206: “constant recruitment”. My approach would be to make the intercept a random effect to vary based on the year of “recruitment” using a Generalized Mixed Effect Model. Then, you do not have to assume constant recruitment although your sample size may not allow you to estimate. It is worth trying.

Line 207: Constant survival + constant natural mortality. Does it imply that there is non-natural mortality? If they only experience natural mortality. If natural mortality is constant, the survival rate must be constant. If there is a way natural mortality is constant but survival can vary (or vice versa), you should explain it. Otherwise, you should eliminate assumption 3.

Reviewers' comments:

Reviewer's Responses to Questions

**Comments to the Author**

1. Is the manuscript technically sound, and do the data support the conclusions?

Reviewer #1: Yes

2. Has the statistical analysis been performed appropriately and rigorously? 

Reviewer #1: Yes

3. Have the authors made all data underlying the findings in their manuscript fully available?

Reviewer #1: Yes

4. Is the manuscript presented in an intelligible fashion and written in standard English?

Reviewer #1: Yes

5. Review Comments to the Author

Reviewer #1: General Comments.

This is a nice paper that presents demographic data for two rare mussel species using conventional mussel survey data. Overall the paper is well written, though tends to drag on in the discussion, and the inferences made by the authors seem plausible based on their findings. I think the paper will be useful for practitioners and therefore should be published. My only real concern is whether issues with sampling from year-to-year (i.e., differences in number of quadrats and no accounting for incomplete detection) is driving some of their results. I don’t think this is a fatal flaw, just something that needs to be discussed. Below are my specific comments and I’m happy to review this paper again, if needed.

Specific comments.

Line 145 to 152 – Unclear how the authors determined the number of quadrats sampled. The authors mention % of area but do not provide % of area per site. The authors then state number of quadrats sampled per site varied from year-to-year and so at least 80 quadrats were excavated at each site to increase precision of estimated densities. However, it’s unclear how they determined this. It seems to me that since data derived from these sampling efforts is being used to make inferences about population condition the authors should then spend some time clearly detailing how mussels were sampled, which should include some information justifying sample sizes so that readers can determine for themselves whether changes in population condition are due to natural processes or sample bias.

Line 164: The authors state that logarithmic transformation of the sample estimate was used to calculate 95% confidence limits if data were non-normally distributed. I find this statement confusing given that in the previous sentence the authors stated that each census sample was evaluated for normality. My understanding is that log transformation is often used to deal with skewed data, which if uncorrected can yield invalid statistical results using parametric statistical tests. However, log transformations only work if the original data follow a log-normal distribution. So, my question is what does 95% CI have to do with addressing non-normally distributed data and did the transformation fix their issue?

Line 182: It would be helpful if the authors provided more details on the age estimates. Simply stating ages of live mussels were estimated using von Bertallanfy growth equations doesn’t tell you anything beyond how age was analytically determined. That is, did the authors thin-section shell to determine age-length relationships or did they use external annuli or some combination of both?

Line 206: I wonder if just truncating age cohorts really addresses violations of these assumptions? For example, do we really know whether recruitment is constant from year-to-year? Or how just omitting size classes less than 1 year old minimizes violation of assumptions 2 and 4? Presumably mortality would be higher in younger cohorts but where you draw the line is anyone’s guess. Same goes for natural mortality. Do the authors have any sense of how violations of these assumptions influence their estimates? It seems to me a more productive way to handle this is to talk about what they did (i.e., omitting smaller size individuals) but then elaborate on how violations of these assumptions affect their inferences. As GEP Box once said, “All models are wrong, but some are useful.” In that spirit, the authors should help clarify the uncertainty about the true parameter values – in this case mortality.

Line 229: Just caught this but the authors switch from using scientific to common names. My recommendation is to use scientific names throughout.

Line 251: The authors may want to consider using IHA (Indicators of Hydrological alteration) to calculate annual flow statistics that may be helpful in explaining year-to-year variation in growth, longevity and survivorship. Rypel et al. (2009) takes this approach to explain differences in mussel growth between regulated vs. unregulated rivers.

https://www.conservationgateway.org/ConservationPractices/Freshwater/EnvironmentalFlows/MethodsandTools/IndicatorsofHydrologicAlteration/Pages/indicators-hydrologic-alt.aspx

Rypel, A.L., W.R. Haag, R.H. Findlay. 2009. Pervasive hydrologic effects on freshwater mussels and riparian trees in southeastern floodplain ecosystems. Wetlands 29: 497-504.

Results – This goes back to my question regarding details on sampling. I wonder how much of the lack of significance reported throughout or even the trends noted by the authors is due to sampling bias. It seems to me that varying level of effort (i.e. different number of quadrats sampled from year-to-year) and no accounting for detection could be driving things. I think the authors need to address this somewhere, maybe in the methods section and then provide a paragraph in the discussion that talks about limitations of their findings and future opportunities to build off this research and validate their findings.

Line 542: This is a great paragraph but I wonder how much of this “sampling variation” is due to differences in effort between years and not accounting for incomplete detection.

Line 625: Bonanza, really? Delete “a bonanza in”

Line 642: The age structure, demographic vital rates and life history section is too long, and it needs to be revised. I’ve read it several times now and I’m still not 100% sure what exactly they are trying to say. My suggestion is for the to introduce life history theory in the beginning and then present their findings so it’s clear which strategy their focal species belong to. While doing this they could then discuss how these traits may explain mussel-environmental relationships observed during their study. They should also talk about how these traits could be used to inform future conservation activities for both species. Winemiller (2005) does a good job doing this but uses fish instead of mussels.

Winemiller K.O. (2005). Life history strategies, population regulation, and implications for fisheries management. Canadian Journal of Fisheries and Aquatic Sciences 62: 872-885.

6. PLOS authors have the option to publish the peer review history of their article (what does this mean?). If published, this will include your full peer review and any attached files.

Reviewer #1: No

---

## [Author Response · Author response to Decision Letter 0]

24 May 2021

Date: April 21, 2021

To: Dr. Masami Fujiwara, Academic Editor, PLOS One

From: Dr. Jess Jones, U.S. Fish and Wildlife Service, Department of Fish And Wildlife Conservation, Virginia Tech, 106a Cheatham Hall, Blacksburg, VA 24061-0321, USA. Phone 1-540-231-2266, Fax 1-540-231-7580, e-mail: Jess_Jones@fws.gov

Subject: Revision of “Long-term Monitoring of Two Endangered Freshwater Mussels (Bivalvia: Unionidae) Reveals How Demographic Vital Rates Are Influenced By Life History Traits of Each Species”

Dear Editor Fujiwara,

We have revised our manuscript based on your comments and those of the referee. We kindly thank you for your feedback and believe that the paper is now greatly improved as a result of your comments to the previous draft. Following this cover letter is a detailed response to each comment. However, I will highlight several items here so you are aware of the major changes to the paper. First, we removed the density dependence analysis from the paper. We ultimately felt that the time-series was not sufficiently long-enough for us to conduct a high quality test of density dependence. Further, the sampling error for one of the species was too high for us to feel confident in this type of analysis. 

As suggested, we used the Indicator of Hydrological Alteration model to examine stream flow and its influence on population growth rate. This analysis and an accompanying Akaike Information Criterion ranking of the respective models is provided. 

We also clarified in more detail how we separated out the sampling and process error. A detailed analysis is provided in the Supporting Information Files (See S3 File). As you suggested, we also ran our data through the Multivariate Autoregressive State-Space Model in the MARSS package in program R to separate the sampling and process error. The results were essentially the same as our current analysis and thus did not tell us anything different. Hence, we have retained our analysis and methodological approach because of its simplicity for characterizing the sampling and process error, and this approach is well documented in the scientific literature. Further, we check our data for autocorrelation using the Durbin-Watson test, which was negative for both species’ time series data. However, if you feel the MARSS model is superior, we are open to replacing this analysis. We can share the MARSS analysis with you if you would like to take a look, just let us know. 

Our base map of the river drainage was made in ARCGIS using a USGS HUC data layer and then modified in Microsoft Power Point to provide sampling locations, which should be ok to publish in PlosOne but let us know otherwise. 

The mussel density data per site in the Clinch River was previously published in peer-reviewed journals in 2014 and 2018, which we cite in the methods section of the paper. However, detailed demographic analyses, which is the focus of our PlosOne submission, were not conducted in those papers. Hence, the data analyses in the current manuscript does not constitute dual publication. Our analyses provided in the current paper are some of the first and most detailed demographic analyses for mussels based on a long-term 10 year time series, which is exceedingly rare for this faunal group in North America. 

We have added co-author Brett Ostby to the paper. He conducted the IHA and AIC analyses, reviewed and help co-write the current draft, and was instrumental in the data collection from 2004-2014. Finally, because of the major revisions to the paper, we are providing two marked-up copies of the manuscript. The first copy shows the major revisions to the paper but because they were so extensive it became difficult to work from that draft. So we created a clean copy of that draft and revised it one last time to create our final clean copy. You can see our final round of revisions on this second marked-up copy. Please let me know if I can be of further assistance during the submission and review of the manuscript.

 Sincerely,

 Jess Jones

E-mail addresses of co-authors:

Tim Lane: tim.lane@dwr.virginia.gov

Brett Ostby: ptychobranchus@gmail.com

Bob Butler: pegias11@gmail.com

Additional Editor Comments

Comment (1) on Density Dependence:

The method for detecting density dependence in the manuscript is not acceptable (lines 433-). 

Please read: Freckleton, R. P., A. R. Watkinson, R. E. Green, and W. J. Sutherland. 2006. Census error and the detection of density dependence. Journal of Animal Ecology 75:837-851

The issue is that the linear regression includes the same variable (with potentially large sampling error) as a response and explanatory variables. You wrote “This relationship was strongest when population density at a site was low, as growth rate tended to increase much more rapidly in the subsequent time interval at each site” (lines 436-438). This is a strong indication that the association is spurious. The only way you can detect density dependence from the type of data you have is to use a state-space method.

Please read: Lebreton, J.-D., and O. Gimenez. 2013. Detecting and estimating density dependence in wildlife populations. The Journal of Wildlife Management 77:12-23.

Authors Response: In review of the two above articles recommended by the editor, we agree that the method used in our original draft submission for density dependence is outdated and may lead to spurious results. Hence, we have removed the density dependence methods, results, figures and discussion from the manuscript. We plan to gather more sampling data and use appropriate methods (state space approach) to test for the potential of density dependence in our study species in a future analysis.

Comment 2 on Separation of Process and Sampling Error (Line 173- ):

“Sampling error variance was determined from the variance of the 11 annual abundance estimates. …” This part does not make any sense to me. You only had 11 samples per cite. If you estimate the variance from 11 samples, it should be the total variance. I am speculating it is something to do with including or excluding spatial variations (among cite variations). It may also be something to do with not understanding the symbol that looks like curly “F” (line 172); it is not defined anywhere. A standard approach to separate sampling error and process error is to use a state-space method. There are many papers on this topic, but I suggest you read the following paper: Holmes, E. E., E. J. Ward, and K. Wills. 2012. MARSS: Multivariate autoregressive state-space models for analyzing time-series data. R Journal 4:11-19.The paper describes a multivariate case, but it is easy to collapse down to a single variable model. The time series do not look stationary. Non-stationarity might inflate process error unless you include other processes (like density dependence or other covariates).

Authors Response: The way we calculated Sampling Variance was by calculating the average variance per site each of the 11 annual abundance sample variances. The total variance per site was the variance of each of the 11 point estimates, themselves. The Temporal variance was, therefore, calculated by subtracting sampling variance from total. The standard deviations would be the sqrt of those variances and the coefficient of variation would be the standard deviation divided by the mean. We calculated the sampling variance and temporal process variation per site and by combining the data from all three sites. Our approach and results for the variance calculations are available in Supplementary Information File S3. We also ran our data through the Multivariate Autoregressive State-Space Model in the MARSS package in program R. The results were essentially the same as our current analysis and did not tell us anything different.

Estimation of Instantaneous Mortality (Line 195-):

Do you use dynamic data or static data? In other words, do you use N_0 when N_t was recruited (dynamics) or N_0 of the same year N_t was estimated (static)? In reality, the cohort that was recruited 2004 (defined as age 0) will appear as age 1 in 2005, age 2 in 2006, etc. It is not clear what is meant by “the number in a year class at time t” (line 199). Which year class? If you took the mean of all densities of the same age, I suggest you separate them. Overall, the analysis is not sufficiently described. Note there are many ways to conduct the analysis, and each has associated assumptions. It is important to clearly describe the method and justify it (this is true with all of the methods you use).

Authors Response: For both species, we used the estimated ages – as determined by a species specific von Bertalanffy equations in Jones et al. (2011) – of all individuals per site and per year (2004-2014) to obtain the frequency of each age class in the river. To estimate mortality, a simple linear regression was conducted on the total age-class frequency data. We have modified our methods to read as: “where N ^t is the total number (i.e., frequency) in an age class at time t obtained from all three sites and 11 census (see Figure 4), and similarly N ^0 is the original number in an age class, and Z is the instantaneous rate of mortality [21].” This method generally is a standard approach in fisheries to estimate mortality from age-class frequency data. The estimate of Z that would be obtained from separating the frequency values per site would be the same, and further, we are not trying to estimate site specific mortality rates, our sample sizes are insufficient to do this, at least for Epioblasma brevidens it is, rather we are estimating a mortality rate for each species throughout the study area. 

Minor Comments:

Line 198: Remove parenthesis around “t”. It looks like Z is a function of t. It should be Z times t. Done, parentheses have been removed.

Line 170: clearly define the curly “F”

Line 188: define lambda_G in words later. Ok, we clarified this section of the methods so λG is now clearly defined. 

Line 189: lambda is the finite annual population growth rate, and r is the instantaneous annual population growth rate. Both are called population growth rates and have the unit of per year. The difference is one is a finite rate, and the other is an instantaneous rate. Ok, we clarified this section of the methods so each growth rate is clearly defined. 

Line 191: why r has a bar, and mu has hat? Remove both bar and hat. You can delete “or mu”. It does not matter what symbols others use as long as you define your parameter clearly. Ok, we removed the bar and hat from the parameters.

Line 192: insert r bar after “the arithmetic mean”. Ok, done.

Line 200: “analogous”. It is a simple linear regression (not analogous). You probably do not need much explanation about the catch-curve analysis. It is a standard method in fisheries. For ecologists, you can just mention it is analogous to life-table analysis. Ok, done, we clarified both points suggested.

Line 201: “A” is the annual finite mortality rate, and “Z” is the annual instantaneous mortality rate. Both are per-capita rates so that they have the unit of per year. Ok, these rates were defined as suggested.

Line 201: Remove round brackets around “x” in the equation. It is interpreted as b is a function of x as it is written now. Ok, done.

Line 206: “constant recruitment”. My approach would be to make the intercept a random effect to vary based on the year of “recruitment” using a Generalized Mixed Effect Model. Then, you do not have to assume constant recruitment although your sample size may not allow you to estimate. It is worth trying. Ok, from a statistical modeling perspective, the Generalized Mixed Effect Model might be a better model choice than SLR but the assumption of constant recruitment is still a more general problem in the estimation of recruitment. Whether a GLMM or SLR is used, recruitment will vary from year to year in most natural populations, and in some cases, this variable recruitment will greatly influence the slope of the regression, and hence the estimate of mortality. The advantage of our study design, is that 11 census years are used to estimate the mean age-class frequencies, hopefully improving the reliability of the mortality estimate. Further, whether GLMM or SLR is used, the slope estimate will be very similar, so either approach is going to yield a similar estimate of mortality. If you would prefer a GLMM we can do that, but it will require us to redo and update the methods and results and some of the figures. For now, we are keeping the SLR until you advise us on your preference. 

Line 207: Constant survival + constant natural mortality. Does it imply that there is non-natural mortality? If they only experience natural mortality. If natural mortality is constant, the survival rate must be constant. If there is a way natural mortality is constant but survival can vary (or vice versa), you should explain it. Otherwise, you should eliminate assumption 3. Yes, your point is well taken, this is a redundant assumption and we have removed assumption 3. Thanks for pointing this out. 

Reviewers' comments:

Reviewer's Responses to Questions

Comments to the Author

1. Is the manuscript technically sound, and do the data support the conclusions?

Reviewer #1: Yes

2. Has the statistical analysis been performed appropriately and rigorously?

Reviewer #1: Yes

3. Have the authors made all data underlying the findings in their manuscript fully available?

Reviewer #1: Yes

4. Is the manuscript presented in an intelligible fashion and written in standard English?

Reviewer #1: Yes

5. Review Comments to the Author

Reviewer #1: General Comments.

This is a nice paper that presents demographic data for two rare mussel species using conventional mussel survey data. Overall the paper is well written, though tends to drag on in the discussion, and the inferences made by the authors seem plausible based on their findings. I think the paper will be useful for practitioners and therefore should be published. My only real concern is whether issues with sampling from year-to-year (i.e., differences in number of quadrats and no accounting for incomplete detection) is driving some of their results. I don’t think this is a fatal flaw, just something that needs to be discussed. Below are my specific comments and I’m happy to review this paper again, if needed.

Specific comments.

Line 145 to 152 – Unclear how the authors determined the number of quadrats sampled. The authors mention % of area but do not provide % of area per site. The authors then state number of quadrats sampled per site varied from year-to-year and so at least 80 quadrats were excavated at each site to increase precision of estimated densities. However, it’s unclear how they determined this. It seems to me that since data derived from these sampling efforts is being used to make inferences about population condition the authors should then spend some time clearly detailing how mussels were sampled, which should include some information justifying sample sizes so that readers can determine for themselves whether changes in population condition are due to natural processes or sample bias.

We made some changes to the Methods better explain the origin of the dataset in the text and in response to the specific critiques regarding dual publication. Our dataset was collected and compiled by a network of scientists and managers that recognized annual quantitative data would be necessary to understand populations of these rare species. We were initially constrained by an ethical concern that oversampling might have deleterious effects on mussels. The sites we visited are among the best remaining in the world and some studies have suggested that sampling might have adverse physiological effects. 

The network of scientists and managers used funding from various sources and help from many institutions. Sampling was initiated by Jess Jones in 2004 and at that time, its use was specific to project goals; it was intended to provide abundance estimates for a discrete time period (2004-2008). Several partners were encouraged by the breadth of information gleamed from that sampling approach and helped continue sampling over the next 7 years. As work proceeded, we gained a better understanding of variation increased sampling to minimize bias and improve detection of rare species. The variation in sampling was a function of practical and ethical constraints and also an example of learning and adapting. More recent data should be considered less biased but not so different that comparison to earlier data is invalid. 

At no point in previous publications was the entire dataset from 2004-2014 examined. Our study specifically focused on these two endangered species with the specific purpose of deriving key demographic parameter estimates. Our analyses are new and far more through than previous attempts that examined the mussel assemblage (30 plus species) as a whole, which has been published elsewhere. Earlier analyses focusing on these species only used data from 2004-2008 and had different objectives and fare different methods. Also novel to this analysis was the examination of flow data. 

Line 164: The authors state that logarithmic transformation of the sample estimate was used to calculate 95% confidence limits if data were non-normally distributed. I find this statement confusing given that in the previous sentence the authors stated that each census sample was evaluated for normality. My understanding is that log transformation is often used to deal with skewed data, which if uncorrected can yield invalid statistical results using parametric statistical tests. However, log transformations only work if the original data follow a log-normal distribution. So, my question is what does 95% CI have to do with addressing non-normally distributed data and did the transformation fix their issue?

In the Methods we modified this sentence to read as “Because all site density was not normally distributed, it was right-skewed due to the quadrat sample data being inflated with zeroes and ones, a logarithmic transformation of the sample estimate was used to calculate 95% confidence limits (CL) if data were non-normally distributed”. So, yes, our data is distributed log-normally, being right skewed, and therefore we used a non-parametric approach to calculate the confidence limits. The data itself was not log-normally transformed. 

Line 182: It would be helpful if the authors provided more details on the age estimates. Simply stating ages of live mussels were estimated using von Bertallanfy growth equations doesn’t tell you anything beyond how age was analytically determined. That is, did the author’s thin-section shell to determine age-length relationships or did they use external annuli or some combination of both? Ok, we fixed this by adding to this sentence so it now reads as “Ages of live mussels were estimated using von Bertallanfy growth equations [19] specific to females and males of each species; the equations and associated shell growth parameters were derived from shell thin-sections to determine age [4].

Line 206: I wonder if just truncating age cohorts really addresses violations of these assumptions? For example, do we really know whether recruitment is constant from year-to-year? Or how just omitting size classes less than 1 year old minimizes violation of assumptions 2 and 4? Presumably mortality would be higher in younger cohorts but where you draw the line is anyone’s guess. Same goes for natural mortality. Do the authors have any sense of how violations of these assumptions influence their estimates? It seems to me a more productive way to handle this is to talk about what they did (i.e., omitting smaller size individuals) but then elaborate on how violations of these assumptions affect their inferences. As GEP Box once said, “All models are wrong, but some are useful.” In that spirit, the authors should help clarify the uncertainty about the true parameter values – in this case mortality. It is well known in the fisheries literature that these assumptions are regularly violated, especially the constant recruitment assumption, which obviously is unrealistic. We have conducted the catch-curve analyses both ways, by removing the young individuals that do not ”recruit” well to the sampling gear because they are too small and harder for surveyor’s to detect as the larger adults, and with the larger adults. So readers have the two estimates to compare, which is a useful approach. But most importantly, our study incorporates 11 censuses of age-size-class frequency data, which allows us to take the mean over a very significant time period to estimate mortality. This approach alleviates many of the concerns of just conducting a catch-curve mortality estimate just based off of one year of data. Hence, our study is robust to many of the assumptions. 

Line 229: Just caught this but the authors switch from using scientific to common names. My recommendation is to use scientific names throughout. Ok, this has been changed as suggested.

Line 251: The authors may want to consider using IHA (Indicators of Hydrological alteration) to calculate annual flow statistics that may be helpful in explaining year-to-year variation in growth, longevity and survivorship. Rypel et al. (2009) takes this approach to explain differences in mussel growth between regulated vs. unregulated rivers.

We used this opportunity to examine flow data using IHA software. Its use was insightful and helped us to correct some issues with flow statistics, including an incorrect assignment of a low-flow phenomena to an interval and an imprecise calculation of extremes. At the same time, it provided many more statistics that had to be examined using a more intensive statistical approach; that being an examination of covariation and comparison of competing hypotheses. IHA also produced a means to examine whether the study period was different or special. Ultimately, IHA application did not alter the conclusion that flow was associated with population growth, and that that variation was overwhelmingly related to juvenile recruitment. IHA also highlighted that study period had a unique period of low flow and paucity of flood events. Beyond that, many statistics and analysis within IHA were intended to compare regulated streams to unaltered system, so those statistics were not used in deference to more intuitive statistics we though were related to life history.

Results – This goes back to my question regarding details on sampling. I wonder how much of the lack of significance reported throughout or even the trends noted by the authors is due to sampling bias. It seems to me that varying level of effort (i.e. different number of quadrats sampled from year-to-year) and no accounting for detection could be driving things. I think the authors need to address this somewhere, maybe in the methods section and then provide a paragraph in the discussion that talks about limitations of their findings and future opportunities to build off this research and validate their findings.

Line 542: This is a great paragraph but I wonder how much of this “sampling variation” is due to differences in effort between years and not accounting for incomplete detection.

A few things here to consider when we are attempting to detect real population trends versus noise. First, quadrats have a high probability of detecting mussels within units, >0.99 for adults, about 0.7 to 0.9 for juveniles depending on their size. Second, these mussels are present at moderate densities for mussels, 0.25 m-2 or even at much greater densities. Thus, increasing our sample size from 60 to 80 to 100 units does not really change our ability to detect the mussels. It would make a big difference if the quadrat sample units had poor detection. So, the variation among years we observed in study is real, even though confidence intervals are wider than we would like. We would probably have to multiply our quadrat sample size by a factor of 10 to get really narrow confidence intervals to define site densities with absolute precision. The problem with such a high quadrat sample size is that we would do significant damage to the habitat and mussels resources by doing such an intensive sampling design.

There is some evidence that sampling affects growth and perhaps even survival of some species. The handing of mussels, while typically not lethal, can stress the mussels and might even be considered harassment. Many of the species we study are endangered and extremely rare, so we cannot put ethics aside in pursuit of mathematical certainty. The data used in this analysis was collected while simultaneously sampling many rare and protected species. As we move forward with management, we either have this information or none at all. Continuing to sample over time will allow us to better distinguish between noise and signal. 

Line 625: Bonanza, really? Delete “a bonanza in” Ok, we have deleted this phrasing so it now reads as “with a large increase in positive population growth”

Line 642: The age structure, demographic vital rates and life history section is too long, and it needs to be revised. I’ve read it several times now and I’m still not 100% sure what exactly they are trying to say. My suggestion is for them to introduce life history theory in the beginning and then present their findings so it’s clear which strategy their focal species belong to. While doing this they could then discuss how these traits may explain mussel-environmental relationships observed during their study. They should also talk about how these traits could be used to inform future conservation activities for both species. Winemiller (2005) does a good job doing this but uses fish instead of mussels.

We have restructured the discussion and reframed the study objectives to help the reader. We have not reworked as specified but found a different way to narrow the scope and focus the reader’s attention to key conclusion and applications of this analysis.

---

## [Editor Report · Decision Letter 1]

2 Jun 2021

PONE-D-20-36478R1

Long-term Monitoring of Two Endangered Freshwater Mussels (Bivalvia: Unionidae) Reveals How Demographic Vital Rates Are Influenced By Life History Traits of Each Species

PLOS ONE

Dear Dr. Jones,

Thank you for submitting your manuscript to PLOS ONE. After careful consideration, we feel that it has merit but does not fully meet PLOS ONE’s publication criteria as it currently stands. Therefore, we invite you to submit a revised version of the manuscript that addresses the points raised during the review process.

Thank you for the extensive revisions. The same reviewer was not available to review the manuscript this time.  Therefore, I evaluated the previous comments and revisions. I felt the revisions were done reasonably by incorporating the comments/suggestions for the most part. I also provide some additional minor comments separately.

We look forward to receiving your revised manuscript.

Kind regards,

Masami Fujiwara, PhD

Academic Editor

PLOS ONE

Journal Requirements:

Additional Editor Comments (if provided):

Some brief discussions on the representativeness of the samples may be useful for the readers. This relates to the first comment provided by the reviewer on the previous version.

Line 179: I am still confused with the log transformation. You wrote that the data include zeros. How did you handle the natural log of 0? Some clarification will be helpful.

Line 207: What distribution was used (Gaussian?) for 95% CI?

Line 222: A major assumption missing is that all age/size classes have the same catchability, which may be violated with your data (see the comment below).

Line 463: Looking at Figure 8, I suspect ages 0-3 probably have lower catchability. If so, those age classes should be eliminated from the catch-curve analysis. Then, Figure 8 should just include the line fitted to data without ages 0-3. If not, further discussion is probably needed.

Line 569: Correlation analysis is fine, but it is better to use GLM or GAM with careful assessments of distribution, residuals, and fit (although I do not insist on this).

Line 672: CMR sampling has been done with freshwater mussels previously by others (I believe). It may be helpful for the readers if the pros and cons of CMR and the catch curve analysis are briefly discussed. It is also useful to mention that the state-space method was applied but it is not shown because most of the variation was from sampling errors.

---

## [Author Response · Author response to Decision Letter 1]

1 Aug 2021

Date: August 1, 2021

To: Dr. Masami Fujiwara, Academic Editor, PLOS One

From: Dr. Jess Jones, U.S. Fish and Wildlife Service, Department of Fish And Wildlife Conservation, Virginia Tech, 106a Cheatham Hall, Blacksburg, VA 24061-0321, USA. Phone 1-540-231-2266, Fax 1-540-231-7580, e-mail: Jess_Jones@fws.gov

Subject: Revision of “Long-term Monitoring of Two Endangered Freshwater Mussels (Bivalvia: Unionidae) Reveals How Demographic Vital Rates Are Influenced By Life History Traits of Each Species”

Dear Editor Fujiwara,

We have revised our manuscript based on your comments that we received back in early June. Again we kindly thank you for your feedback and believe that the paper now meets with the publication criteria of PlosOne. Following this cover letter is a detailed response to each comment. We are providing a marked-up copy of the manuscript and a final clean copy. Please let me know if I can be of further assistance completing and final revisions to the manuscript.

 Sincerely,

 Jess Jones

E-mail addresses of co-authors:

Tim Lane: tim.lane@dwr.virginia.gov

Brett Ostby: ptychobranchus@gmail.com

Bob Butler: pegias11@gmail.com

PONE-D-20-36478R1

Long-term Monitoring of Two Endangered Freshwater Mussels (Bivalvia: Unionidae) Reveals How Demographic Vital Rates Are Influenced By Life History Traits of Each Species

PLOS ONE

Journal Requirements:

RESPONSE: The references have been checked and all citations in the text are included in the reference list, no additional citations have been added and none have been removed from the previous version.

Additional Editor Comments (if provided):

Some brief discussions on the representativeness of the samples may be useful for the readers. This relates to the first comment provided by the reviewer on the previous version.

RESPONSE: Lines 163-167 have been added in the Methods describing the representativeness of the samples. 

Line 179: I am still confused with the log transformation. You wrote that the data include zeros. How did you handle the natural log of 0? Some clarification will be helpful.

RESPONSE: Since our data was right skewed, we used the method of Strayer and Smith 2003 (Citation #16) for calculating the 95% confidence limits (CL’s). The method uses a logarithmic transformation of the site mean density (D) and a delta-method approximation of the variance to estimate the CL’s. So quadrat samples with no mussels, i.e., 0 values were not transformed, only the site mean and variance of D were transformed to obtain the CL’s. The method has been applied widely to quadrat sampling of freshwater mussel communities to estimate CL’s. We have updated this part of the sentence on line 179 to make it more specific, inserted the formula we used and added the appropriate citation in case readers are interested in this method for calculating CL‘s when the underlying quadrat data are right skewed. 

Line 207: What distribution was used (Gaussian?) for 95% CI?

RESPONSE: We assumed a normal (Gaussian) distribution to compute the variance and standard error for each species’ respective population growth rate. However, we did not compute the 95% CI’s for this parameter. 

Line 222: A major assumption missing is that all age/size classes have the same catchability, which may be violated with your data (see the comment below).

Line 463: Looking at Figure 8, I suspect ages 0-3 probably have lower catchability. If so, those age classes should be eliminated from the catch-curve analysis. Then, Figure 8 should just include the line fitted to data without ages 0-3. If not, further discussion is probably needed.

RESPONSE: It is important that Ages 1-3 remain in our analyses, as shown in Figure 8 and as reported in the Results. I’ve updated the figure legend to make it clear that estimates of mortality are available with and without Ages 1-3 included in the analyses. Our analyses do not include Age 0 individuals but our original graphs do extend to the y-axis, giving the impression that Age-0 individuals were included in the analysis. So we have remade the graph to make it clear that Age 0 individuals were not included in the analysis. Generally, we believe that Ages 1-3 have equal catchability relative to the older larger mussels but perhaps lower mortality but that as individuals mature and age their mortality increases, so having mortality rate estimates with and without Ages 1-3 is important for readers to see how the mortality rate will likely increase in the older adult age classes.

Line 569: Correlation analysis is fine, but it is better to use GLM or GAM with careful assessments of distribution, residuals, and fit (although I do not insist on this).

RESPONSE: We used a two-step process for developing competing linear regressions (a form of GLM). Correlation analysis was presented as a screening analysis to inform ultimate linear regression models (our hypotheses). We had many possible independent variables and few independent replicates of dependent variables. Many independent variables were correlated. We used correlation analysis to better inform which of many competing independent variables to include in final model runs (informed hypotheses). To a more limited degree, we did this for dependent variables. We chose only one or two of correlating independent variables that we felt were possibly biologically significant for final models. Final linear regression models were GLMs. GLMs are generally robust to problems with distribution and were used because were looking for linear relationships. We did not think other distributions were appropriate to explore at this time. We used AICc to compare relative support, so fit statistics, such as r2, are not appropriate. As we collect more data at these sites we hope to build and compare other models. GAM may be more appropriate at a later point when we have more than 10 independent replicates. At this point there is no advantage to use GAM over information theory approach to compare competing linear regression models. 

Line 672: CMR sampling has been done with freshwater mussels previously by others (I believe). It may be helpful for the readers if the pros and cons of CMR and the catch curve analysis are briefly discussed. It is also useful to mention that the state-space method was applied but it is not shown because most of the variation was from sampling errors.

RESPONSE: CMR sampling is increasingly popular and authors have been using this approach at other sites and with other mussel species. Two of the coauthors were also coauthors of Carey et al. 2019--a study that compared CMR techniques to quadrat monitoring. We have decided not to go into a discussion of the pros and cons of CMR because it would be lengthy and deserves a standalone review that would expand on the work of Carey et al. 2019. Moreover, CMR would be no more effective due to relative rarity of E. brevidens, which is distributed at low levels across large sties. Seconding, applying CMR at these sites would mean we could not collect data on the dozens of other species we monitor at these sites. Finally, CMR approaches at these sites may put too much pressure on sensitive areas where we limit our impact. Previous CMR studies on the Clinch River required 5 sampling occasions each year for 2 years (Carey et al. 2019) and only produced limited information about those 2 years. As even short-lived mussels live 5 yrs or more, understanding population dynamics over longer time spans is critical.

---

## [Editor Report · Decision Letter 2]

4 Aug 2021

Long-term Monitoring of Two Endangered Freshwater Mussels (Bivalvia: Unionidae) Reveals How Demographic Vital Rates Are Influenced By Life History Traits of Each Species

PONE-D-20-36478R2

Dear Dr. Jones,

We’re pleased to inform you that your manuscript has been judged scientifically suitable for publication and will be formally accepted for publication once it meets all outstanding technical requirements.

Kind regards,

Masami Fujiwara, PhD

Academic Editor

PLOS ONE
---

## [Editor Report · Acceptance letter]

17 Aug 2021

PONE-D-20-36478R2 

Long-term Monitoring of Two Endangered Freshwater Mussels (Bivalvia: Unionidae) Reveals How Demographic Vital Rates Are Influenced By Species Life History Traits 

Dear Dr. Jones:

I'm pleased to inform you that your manuscript has been deemed suitable for publication in PLOS ONE. Congratulations! Your manuscript is now with our production department. 

Kind regards, 

on behalf of

Dr. Masami Fujiwara 

Academic Editor

PLOS ONE